# Estimating neuronal firing density: A quantitative analysis of firing rate map algorithms

**Roddy M. Grieves** *

Department of Psychological and Brain Sciences, Dartmouth College, Hanover, New Hampshire, United States of America

* roddy.m.grieves@dartmouth.edu

## Abstract

The analysis of neurons that exhibit receptive fields dependent on an organism's spatial location, such as grid, place or boundary cells typically begins by mapping their activity in space using firing rate maps. However, mapping approaches are varied and depend on multiple tuning parameters that are usually chosen qualitatively by the experimenter and thus vary significantly across studies. Small changes in parameters such as these can impact results significantly, yet, to date a quantitative investigation of firing rate maps has not been attempted. Using simulated datasets, we examined how tuning parameters, recording duration and firing field size affect the accuracy of spatial maps generated using the most widely used approaches. For each approach we found a clear subset of parameters which yielded low-error firing rate maps and isolated the parameters yielding 1) the least error possible and 2) the Pareto-optimal parameter set which balanced error, computation time, place field detection accuracy and the extrapolation of missing values. Smoothed bivariate histograms and averaged shifted histograms were consistently associated with the fastest computation times while still providing accurate maps. Adaptive smoothing and binning approaches were found to compensate for low positional sampling the most effectively. Kernel smoothed density estimation also compensated for low sampling well and resulted in accurate maps, but it was also among the slowest methods tested. Overall, the bivariate histogram, coupled with spatial smoothing, is likely the most desirable method in the majority of cases.

## Author summary

Spatially modulated neurons in the brain increase their activity when an animal visits specific regions of its environment. Studying these neurons often begins with the creation of a firing rate map: a statistical representation of the cell's activity in space. Different methods, relying on different parameters, are commonly used to generate these maps. These parameters can have a huge impact on the maps created and, in turn, any results derived from them. Yet, very little quantification of these parameters has been attempted and they are almost universally chosen based on qualitative, study-dependent assessments. In this paper we quantify the 'best' combinations of parameters for each method and provide a

**Data Availability Statement:** All code used for running simulations, analysis, and plotting is available in the GitHub repository at https://github.com/RoddyMGrieves/Rate-map-quantification. Code to generate firing rate maps using any of the

methods described in this manuscript is available in the GitHub repository at https://github.com/RoddyMGrieves/rate_mapper.

**Funding:** The author(s) received no specific funding for this work.

**Competing interests:** The authors have declared that no competing interests exist.

way for researchers to calculate these for their own data. Using parameters that are both consistent across studies and quantifiably demonstrated to be the most accurate will reduce the variability between future studies while improving the validity of their results.

## Introduction

Finding food, shelter, mates, and safety all depend on accurate spatial navigation. It is unsurprising then that many regions of the brain harbor spatially modulated neurons. Typically these cells are inactive, exhibiting brief periods of activity in spatially constrained locations [1,2]. Place cells in the hippocampus, for instance, increase their activity in one or more regions known as 'place fields' [3–5]. Grid cells in the medial entorhinal cortex (mEC) exhibit similar firing fields but these are more numerous and span the environment in a triangular grid [6,7]. Boundary cells found in the subiculum and mEC, as well as a number of other regions, have elongated firing fields which extend parallel to environment boundaries [8–10].

The analysis of spatial neurons such as these typically begins by mapping their spatial activity. However, the methods for creating firing rate maps are varied, fragmented, and their performance has not been systematically compared. Additionally, the parameters used to generate firing rate maps are often chosen subjectively and vary greatly between studies. It is unknown how much variability in spatial navigation research can be attributed to these differing approaches, although small changes to analyses such as these have been shown to impact results significantly [11].

Firing rate maps are dependent on two types of multi-dimensional data: position sampling and spike locations. Many advances have been made towards multivariate histogram optimization, for instance, Sturge's rule [12], Scott's rule [13] and the Freedman-Diaconis rule [14] can be applied to estimate how many bins to use when generating a univariate histogram. However, these methods are almost universally adapted for normally distributed data sets [12,15] and position sampling is rarely normally distributed. To the contrary, in open field environments researchers are typically working to obtain a uniform distribution–equal sampling throughout the environment [16]. Conversely, in alleyway mazes coverage is restricted to the shape of the maze and thus position sampling is highly non-normal, and these rules-of-thumb should not be expected to perform well. Additionally, while the spikes contained within an individual place field are often conceptualized or modelled as Gaussian [17,18], they are rarely normally distributed [19–21]. Furthermore, cells often exhibit multiple, possibly overlapping fields [16,22,23] indicating that we can expect spike data to deviate significantly from a normal distribution in the majority of cases as well. Thus, firing rate maps are the result of two very different non-normal distributions, neither of which can be easily estimated or described parametrically.

In the face of these shortcomings, researchers in the field of spatial navigation have persevered using firing rate map approaches and parameters based on subjective assessments, without quantification. To begin to address this issue and start to provide some guidelines which may be used in future, we utilized an empirical approach to compare the effects of different mapping parameters on different mapping methods with the aim of discovering the best parameters to use in each case. The mapping methods we compare here are the bivariate histogram, averaged shifted histogram, adaptive smoothing and adaptive binning, and two kernel smoothed density estimate approaches. The 'best' parameters can take two forms: a) they minimize the error between the firing rate map $r(x,y)$ and the true underlying spike probability distribution $f(x,y)$, thus capturing the fine and large-scale properties of place fields whilst also

compensating for inhomogeneous sampling or, b) they minimize this error in a way that also provides an efficient trade-off with computational speed. This latter factor is important because although individual firing rate maps can be easily and quickly generated using modern computers, more and more analyses are being developed which depend on generating rate maps *en masse* such as bootstrapping [24], spike-train shuffles [25] and cross-validation [26] that benefit from increased computational speed.

Here, we provide equations for each mapping approach that can be used to calculate the best mapping parameters under most circumstances. Comparing across approaches we conclude that smoothed bivariate histograms are consistently associated with fast computation times while providing surprisingly accurate maps, for this reason it is likely to be the preferred method in the majority of cases. As expected, adaptive smoothing and binning approaches compensate for low sampling the most effectively, but among these adaptive binning demonstrates the highest accuracy across a wide range of parameters while balancing computation time and other factors. Kernel smoothed density estimation is among the slowest methods but exhibits consistently low errors and demonstrates the highest accuracy when computation time is unlimited. In the literature, the mapping parameters commonly reported for the histogram, KSDE and adaptive smoothing methods (the most widely used) closely resemble the parameters our analyses highlight as Pareto-optimal, this suggests that researchers can make, and generally have made, good qualitative assessments of optimal map accuracy.

## Results and discussion

When we create a firing rate map using real-world data, we have no way to quantify its accuracy because we do not know the true underlying activity pattern of the cell. To solve this issue, we used a simulated dataset which mimicked as closely as possible the positional trajectories of rodents and the spatial properties of hippocampal place cells. This allowed us to construct firing rate maps and quantify their accuracy by comparing them to the simulated cell's ground-truth firing distribution. We will start by providing a brief overview of this approach, we will then briefly describe each mapping method, greater details for which can be found in the corresponding methods sections. We then quantify the error associated with each mapping method, its optimal parameters and the effect of recording duration and field size on these. Lastly, we compare the different methods and compare our results to the parameters most commonly reported in the literature.

### Simulated dataset

First, we simulated random walk trajectories which mimicked a rat exploring a 1.2 m × 1.2 m square environment (Methods: *Random walk*). These walks simulated 64-minute-long sessions (Fig 1A and 1E) which were also clipped to 4- and 16-minute durations so that we could investigate the effects of recording duration and behavioral coverage. We generated 8 unique trajectories of each duration to reflect the fact that place cell datasets normally include multiple sessions from different rats.

We then simulated the spike probability distributions of three groups of 256 place cells; the three groups differed in their average field diameter (Fig 1B and 1D; Methods: *Place cells*) so that we could investigate the effects of spatial selectivity. Using the random walk trajectories and a Poisson function to generate spikes, we transformed these spike probability distributions into simulated spiking data (**Fig 1**C; Methods: *Spiking*). Spikes were randomly shifted in time by small amounts to mimic errors in behavioral tracking and oscillatory dynamics such as phase precession. Cells were also simulated to exhibit different average firing rates and a small amount of random out-of-field activity.

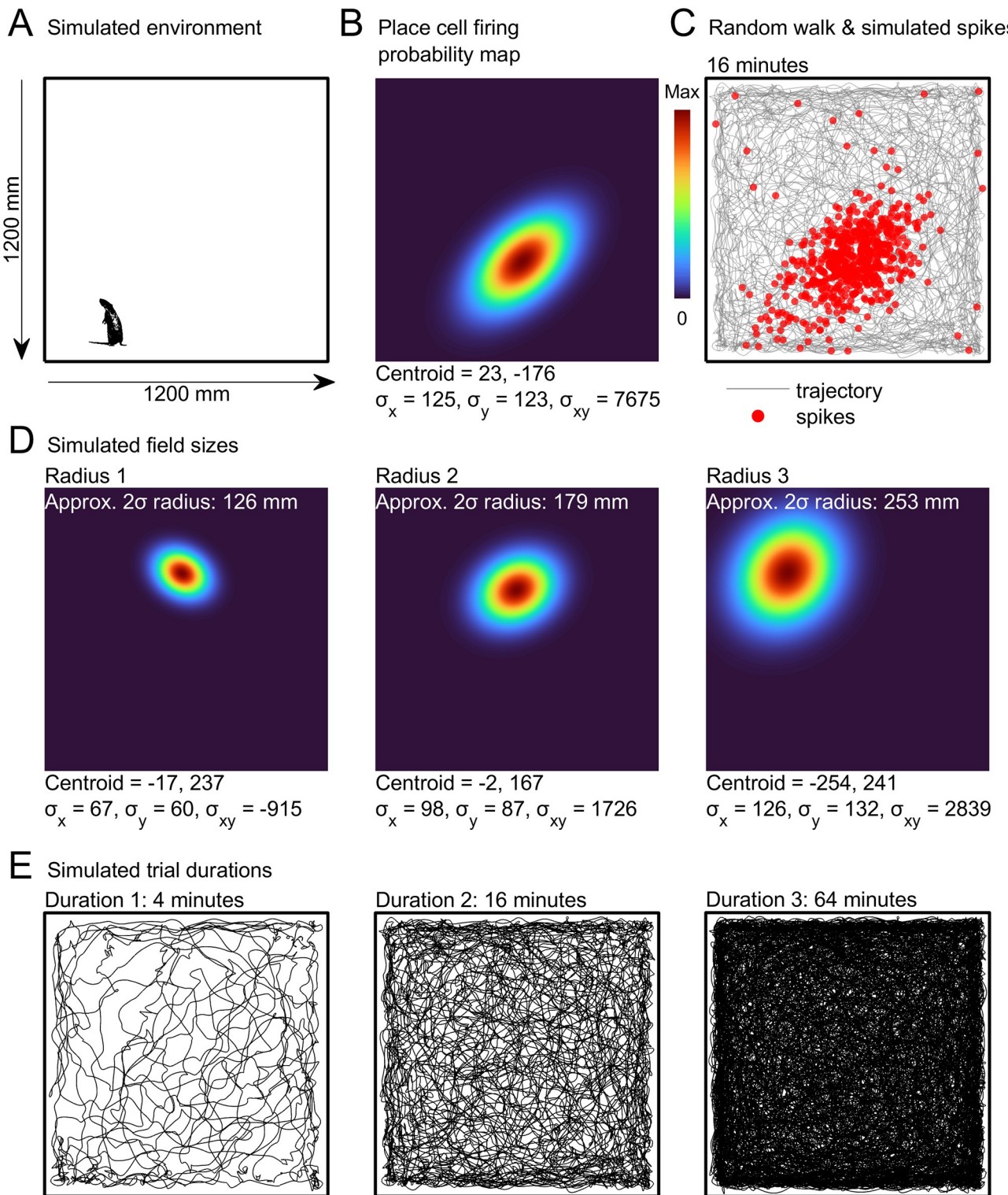

**Fig 1. Trajectory and place cell simulations. A)** The simulated environment with a scale rat for comparison. **B)** The spike probability map for a simulated place cell with a single firing field. The field parameters are given below: the centroid is the center point of the field, the coordinates are relative to the center of the map, $\sigma_x$ and $\sigma_y$ give the field's standard deviation in x or y respectively, $\sigma_{xy}$ gives the field's variance in x and y and is used here to give fields an orientation oblique to the x- and y-axes. **C)** Place cell spikes, simulated using the probability map in b and a random walk trajectory. **D)** We simulated place cells with three different firing field sizes, examples exhibiting a single firing field are shown from each group, text above gives the average field radius of the group and the field parameters are given below as in b. **E)** We simulated 8 different random walk trajectories,

these were clipped to a 4-, 16- or 64-minute duration. Shown here is one trajectory clipped to these three durations (for the kernel smoothed density estimate a 24-minute duration was used in place of the 64-minute one).

Next, for each combination of recording duration and firing field radius we generated firing rate maps for all 256 place cells using each mapping method while varying their main parameters, such as bin size and smoothing strength. We then compared these maps to the ground-truth spike probability maps using mean integrated squared error (MISE; Methods: *Map accuracy*). Lastly, we found the combination of parameters which either a) minimized MISE, thus providing the most accurate map possible or b) provided a Pareto-optimal solution [27]: simultaneously minimizing MISE, computation time and the proportion of empty map bins while maximizing the accuracy with which place fields could be detected (Methods: *Parameter optimization*). The MATLAB code used to generate each type of firing rate map is provided alongside this paper.

### Generating firing rate maps

**Histogram.**  The earliest, simplest, and most widely used method for constructing firing rate maps is to bin the spike and position data into a grid of non-overlapping square pixels. The resulting bivariate histogram maps crosstabulations of values in *x* and *y*. This method was first proposed as a firing rate map approach by O'Keefe [28] and later by Muller, Kubie, and Ranck [4] (linear firing rate maps were also proposed around this time by Barnes et al. [29] and McNaughton et al. [30]). By dividing a histogram generated using the spike *xy* coordinates by a histogram generated using position *xy* coordinates (multiplied by the sampling interval of this data) a third histogram is obtained which gives the firing rate, in Hz, of the cell in every bin. Often, the spike map and dwell map are also smoothed before this division (or very rarely, after), resulting in a smoothed firing rate map. This process can be seen in **Fig 2** and is formalized in Methods: *Histogram*.

Histograms depend on two input parameters, the bin size (i.e., the distance between neighboring pixels of the histogram) and, if used, the smoothing strength, typically expressed as the standard deviation of the smoothing kernel if it is Gaussian. Initially, histogram firing rate maps were generated using large bin sizes, usually dictated by the resolution of the camera used to track the animal and were left unsmoothed. However, more recent approaches tend to utilize smaller bin sizes combined with Gaussian smoothing (see section *Comparison to the literature* below). We generated histogram firing rate maps for our simulated place cells using bin sizes ranging from 1 to 640 mm and a Gaussian smoothing kernel with standard deviations ranging from 0 (no smoothing) to 640 mm (Methods: *Histogram*). These ranges of values were chosen so that they encompass the values reported in the literature with a significant margin.

Should firing rate maps be smoothed before or after division? In the vast majority of cases spike and dwell time maps are smoothed and then used to produce a firing rate map which does not require its own smoothing. Alternatively, unsmoothed spike and dwell time maps can be used to produce a firing rate map which then has smoothing applied to it. The former approach benefits from the fact that the original spike and dwell maps contain no missing values (empty bins simply equal zero), which makes smoothing with a weighted kernel trivial. By comparison, firing rate maps do contain missing, or *NaN (Not a Number)*, values in positions where the dwell time was equal to zero. This complicates smoothing because the natural tendency of a smoothing kernel is to propagate missing values—when a single *NaN* falls within the kernel window the returned result will be *NaN*. However, more complex kernels can be constructed which ignore these missing values and smooth only the numerical data [31]. We briefly investigated the implications of this effect and found that, generally, smoothing after

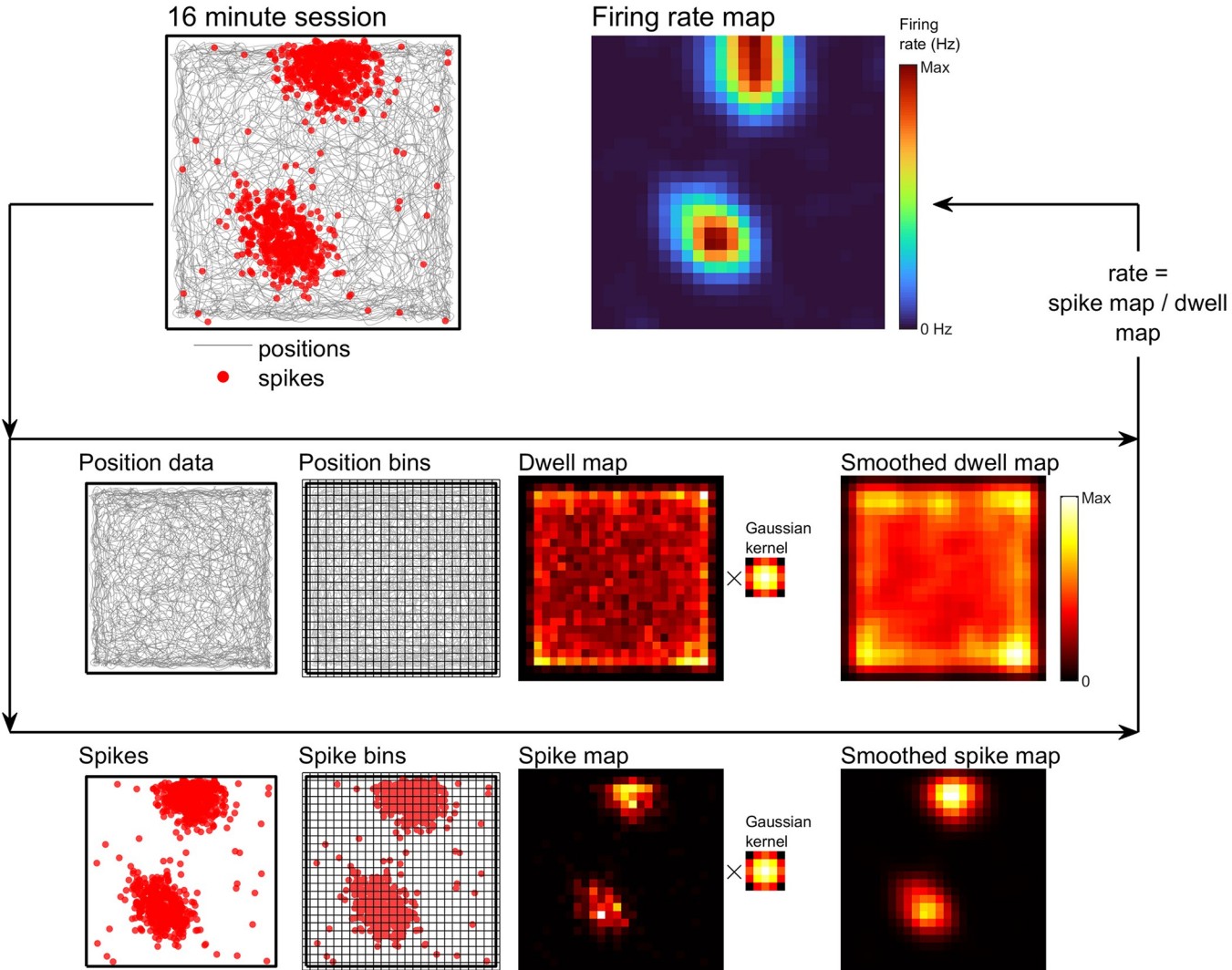

**Fig 2. Schematic showing the histogram method.** See Methods: *Histogram* for more detail. Schematics showing the steps involved with creating a histogram firing rate map. Position and spike data are binned separately into unsmoothed maps. Position counts are multiplied by the sampling interval of the position data to obtain the dwell map. For this example, 50 mm² bins are used. These maps are then, optionally, smoothed with a Gaussian kernel, in this case a 5×5 bin kernel with the standard deviation set to 50 mm, to produce a smoothed dwell and spike map respectively. The spike map is then divided by the dwell map to obtain a map of average firing rate.

division does not greatly improve accuracy (**S1 Fig**), thus, we smoothed our maps only before division.

One potential disadvantage of smoothing before division is that it can introduce fictional values in bins that are empty by propagating values from nearby visited bins. By comparison, smoothing after division assigns firing rate values only to bins that contained at least one position data sample, because there is no such propagation of values in the dwell map. The result is a map which more honestly represents the position sampling and may be more valuable in complex, alleyway mazes with consistent trajectories. Alternatively, this propagation problem can also be solved by imposing a minimum dwell time on bin values, thus removing the small dwell times resulting from smoothing [32], by setting bins as empty if the closest position data sample is more than some cut-off distance away [33], or by emptying unvisited bins after smoothing [34].

**Averaged shifted histogram (ASH).** One issue that has been raised with histograms is that the locations of the bins are somewhat arbitrary [35]. It is often possible to shift the edges of the bins in one or both dimensions while still enclosing all of the data, and in some cases this can lead to very different final distributions [36,37]. This issue led Scott [35] to propose the averaged shifted histogram (ASH) [35,38,39]. The ASH utilizes the histogram procedure outlined above (**Fig 2**) but instead of one histogram and one set of bins, numerous histograms are generated with slightly offset bin edges. The final ASH is then calculated as the average of these offset histograms (Methods: *Averaged shifted histogram*). This approach is computationally very efficient but typically benefits from a significant decrease of error over the traditional histogram [35,36,38] and often only requires around 80% of the data [35]. Firing rate maps can be calculated by dividing a spike map ASH with a dwell map ASH as with the histogram method.

The averaged shifted histogram depends on two input parameters: a bin size which acts as in an ordinary histogram and a smoothing parameter $m$, which defines how many sub-histograms are generated and thus, the offset between histograms, δ, which is equal to the bin size / $m$. If $m$ is set to 1 the ASH closely resembles the result of a bivariate histogram with no smoothing. Conversely, as $m$ approaches infinity (or the maximum resolution of the data) the ASH approximates a kernel smoothed density estimator [35,38]. Thus, we generated firing rate maps using bin sizes ranging from 1 to 640 mm and smoothing $m$ values ranging from 2 to 64 (Methods: *Averaged shifted histogram*).

One benefit of this method is the increased resolution of the resulting maps: a histogram and an ASH rate map generated with a bin size of 20 mm will look qualitatively similar, but if the ASH map is generated with a smoothing $m$ of 5, for example, it will have 25× greater resolution owing to its subdivided bins. This increased precision could make ASH maps valuable when quantifying place field spatial properties such as their shape, area, or centroid as it will allow more precise measurements. Despite these advantages, as far as we are aware this approach has never been implemented in the spatial navigation literature.

**Adaptive smoothing.** Smoothing provides a convenient way for bins to make use of surrounding data and interpolate across regions with low or missing sampling. Ideally though, each bin would 'adapt' to the density of the surrounding data, enlarging when there is not a great deal of local information, but shrinking in regions where the local data density is very high [40]. An approach like this for firing rate maps was first described by Jung et al. [41] but was only fully described later by Skaggs and McNaughton [42] which is the approach we have adopted here. A similar, but less frequently adopted variant was described by Skaggs et al. [43] (see Methods: *Adaptive smoothing*) and is not discussed here. This 'adaptive smoothing' allows bins to differ in size, depending on the density of the surrounding data (**Fig 3**). The aim is to allow bins which contain too little sampling to expand until they contain a minimum number of spikes and position data. The adaptive nature of this approach means that every firing rate map bin results from a trade-off between smoothing error (data are too blurred together) and sampling error (too few samples to estimate firing rate accurately). The pixelwise approach described by Skaggs and McNaughton [42] can be seen in **Fig 3**. However, as discussed in Methods: *Adaptive smoothing*, this method can be accelerated using pre-binning and convolution, as shown in **Fig 4**. This accelerated process generates virtually identical firing rate maps, but in a small fraction of the time (**S2 Fig**) and so this was the method we employed here.

Adaptive smoothing depends on two input parameters: an initial bin size which sets the minimum area a bin must occupy (this ensures all of the data is included in at least one bin) and a tunable smoothing parameter $\alpha$ which controls the minimum number of spikes and position data required within each bin. As before, we used bin sizes ranging from 1 to 640 mm

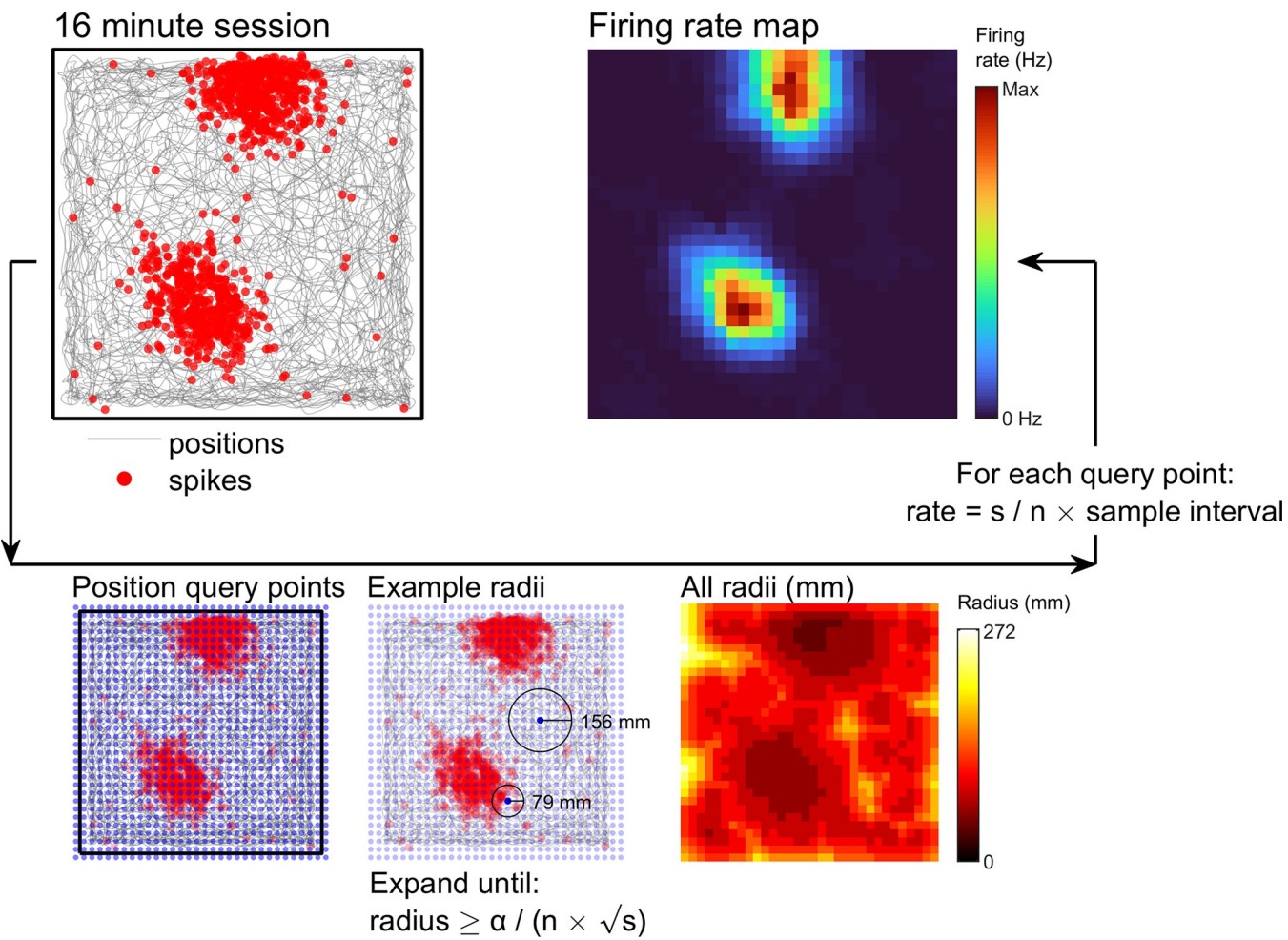

**Fig 3. Adaptive smoothing, pixelwise method.** See Methods: *Adaptive smoothing* for more detail. Schematic showing the steps involved with creating a firing rate map using the adaptive smoothing approach described by Skaggs and McNaughton [42]. A set of query points are specified, these typically form a square grid spanning the data. A circle is expanded around each point until the contents satisfy the adaptive equation (bottom of figure). The radius needed to satisfy the adaptive equation is shown for two query points. The value of the firing rate map at each query point is then equal to the number of spikes divided by the number of position samples within that circle multiplied by the sampling interval of the position data.

and to capture the full range of smoothing α values reported in the literature we tested values ranging from 100 to 32000 (Methods: *Adaptive smoothing*).

## Adaptive binning

A similar method to Skaggs and McNaughton's [42] adaptive smoothing was proposed by Yartsev and Ulanovsky [44] for data collected from freely flying bats. A variation of this was also used by Ginosar et al. [45]. In the Yartsev and Ulanovsky [44] approach, bins are expanded until they contain a minimum 1 second of recording data, regardless of the number of spikes. The firing rate in a bin is then equal to the number of spikes within this radius of the bin divided by the total time spent within this radius. This method is similar to adaptive smoothing (**Fig 3**) but instead represents K-nearest neighbor density estimation [40], where the 1 s time interval can also be understood as K equal to the position sample rate. Compared to adaptive smoothing this method places emphasis on behavioral sampling rather than a combination of sampling and spike count. Like adaptive smoothing, this method can be accelerated

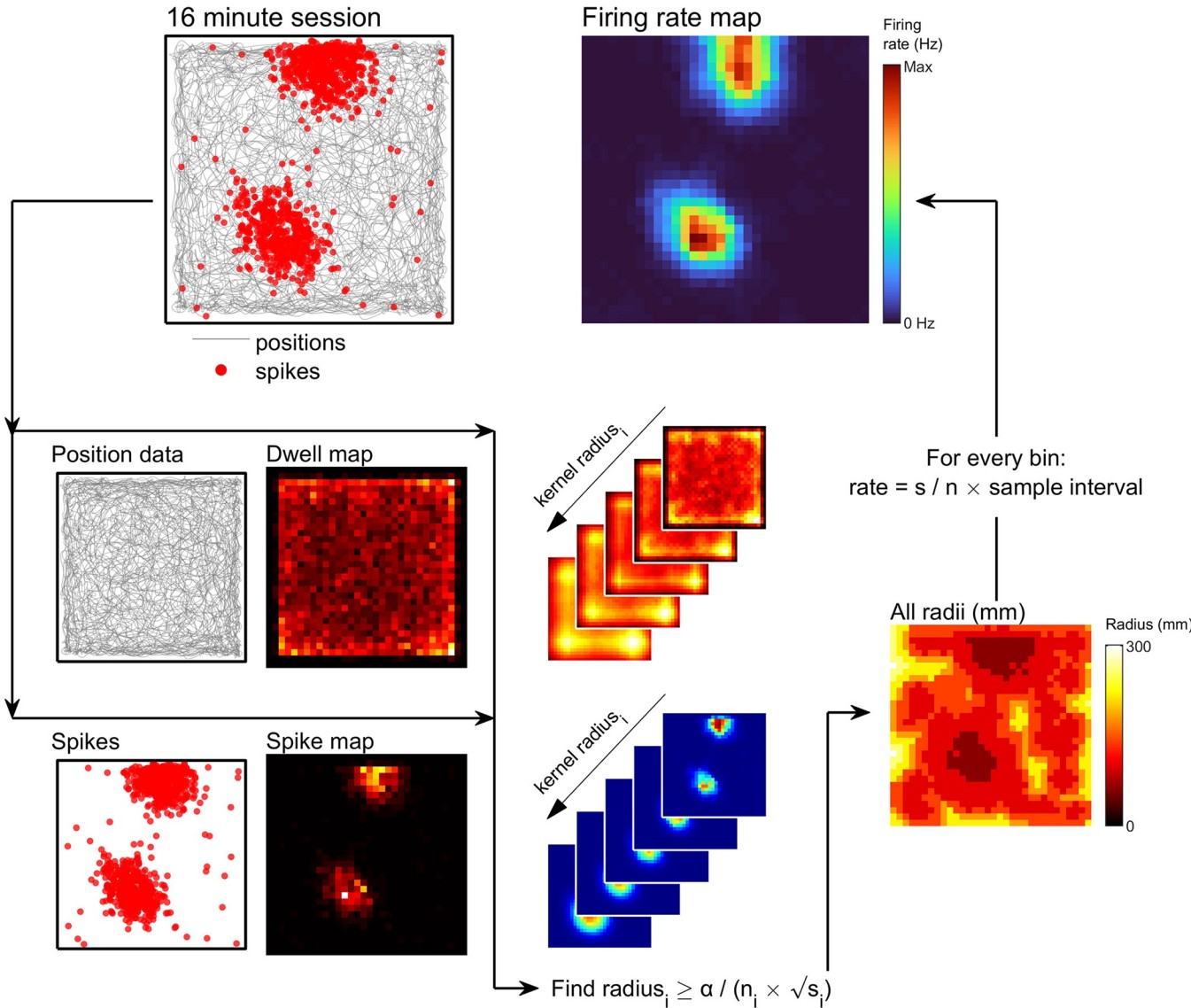

**Fig 4. Adaptive smoothing, convolution method.** See Methods: *Adaptive smoothing* for more detail. Schematic showing the steps involved with creating a firing rate map using the adaptive smoothing approach accelerated through convolution. Position and spike data are binned separately into unsmoothed dwell and spike maps respectively. For this example, 50 mm bins are used. Convolution with circular unity-gain kernels of varying sizes is used to sum the total spikes and position samples at a set of discretized radii. For each bin, the smallest radius of kernel at which the adaptive equation can be satisfied is then found. The value of the firing rate map in that bin is equal to the number of spikes divided by the number of position samples within the kernel multiplied by the sampling interval of the position data. The result is a map which is functionally identical to one generated using the pixelwise approach (**Fig 3**) but in a fraction of the time (**S2 Fig**).

using a convolution-based procedure (similar to that in **Figs 4** and **S2**). Here, we used a two-dimensional implementation of the method proposed by Yartsev and Ulanovsky [44], accelerated in this way.

Adaptive binning depends on two input parameters: the first is the initial bin size which sets the minimum area a bin must occupy (this ensures all of the data is included in at least one bin) the second smoothing parameter *t* defines the minimum enclosed duration at which bin expansion is halted. Yartsev and Ulanovsky [44] used 10 cm³ voxels and set *t* to 1 second, so we generated firing rate maps using two-dimensional bin sizes ranging from 1 to 640 mm and smoothing *t* values ranging from 0.5 to 10 seconds (Methods: *Adaptive binning*).

## Kernel smoothed density estimate (KSDE)

Both the bivariate histogram and adaptive methods estimate data density through binning fine-scale data points into, somewhat arbitrary, discretized portions of space. Smoothing allows data points to affect bins other than the one they are assigned to, but the resulting estimate is always discontinuous–values are only known for each bin location and each bin contains only one value. Adaptive smoothing and binning further extend the histogram method, allowing more data to be utilized in under-sampled regions. However, the kernel smoothed density estimate (KSDE) proposed by Rosenblatt [46] and Parzen [47], also known as the Parzen–Rosenblatt window method, takes this a step further and estimates local data density based on all of the available data, producing a continuous estimate which can be queried at any location.

Although KSDE's are often considered one of the most accurate approaches for data density estimation [48,49], they are not commonly utilized to generate firing rate maps. Leutgeb et al. [50,51] described what has become the most widely adopted approach [7,9,33,52–55], by estimating firing rate as the ratio of two kernel smoothed density estimates, one calculated on position data and the other on spikes (Methods: *Kernel smoothed density estimate*; **Fig 5**). In

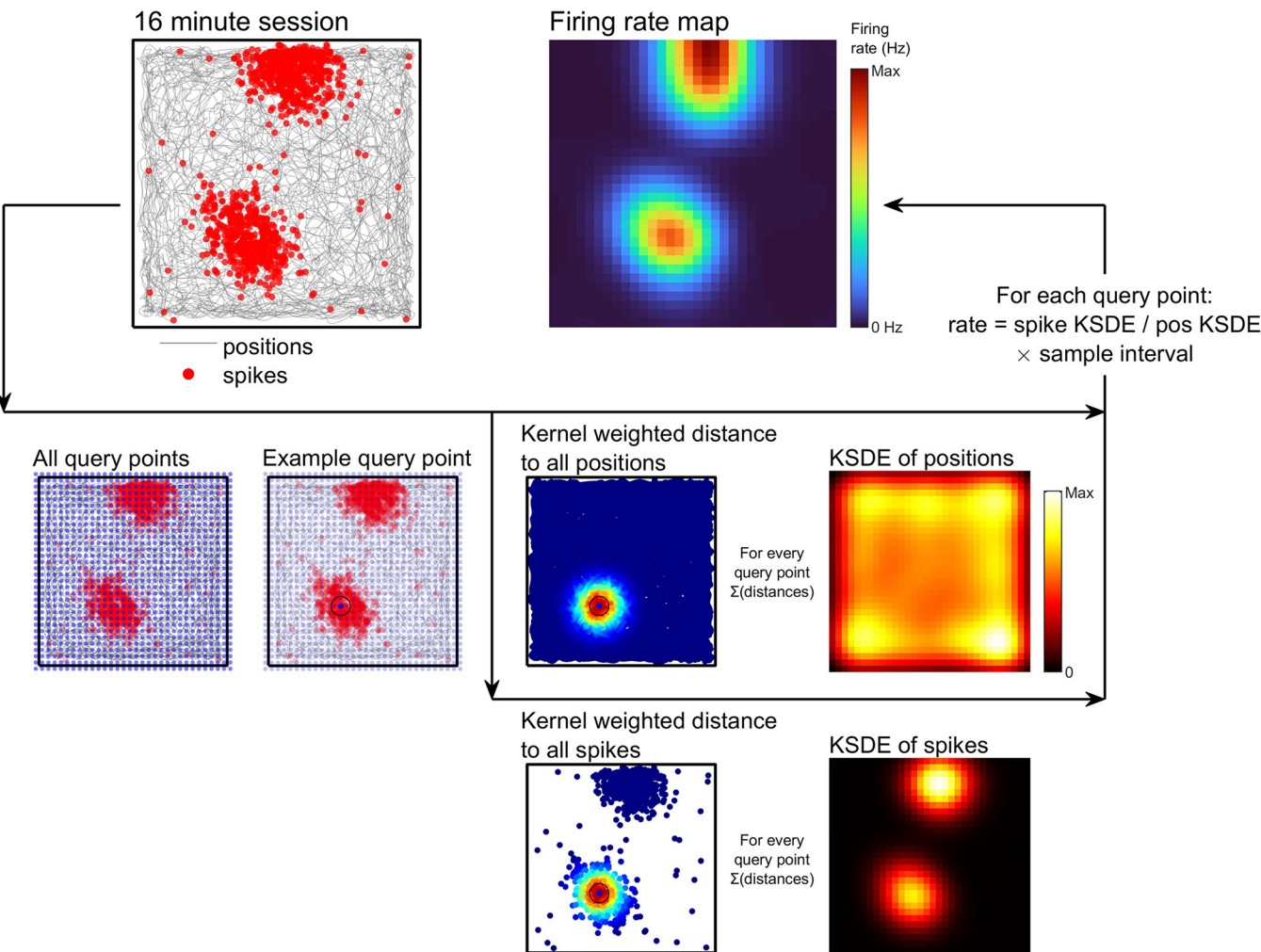

**Fig 5. KSDE firing rate map method.** See Methods: *Kernel smoothed density estimate* for more detail. Schematics showing the steps involved with creating a kernel smoothed density estimate (KSDE) firing rate map. A set of query points are specified forming a square grid spanning the data. For each query point the distance to every position data sample and spike is calculated, these are then kernel weighted and summed. The value of the firing rate map at that query point is then equal to the spike-distance-sum divided by the position-distance-sum multiplied by the sampling interval of the position data.

comparison to the Leutgeb et al. [50,51] approach we found that the MATLAB kernel density estimate implementation (MATLAB function *mvksdensity*), which uses a very similar smoothing kernel, provides a significant speed increase, generates virtually identical firing rate maps (**S3 Fig**) and offers many convenient additions such as boundary correction. Alternatively, combining the two approaches by employing the Leutgeb et al. [50,51] kernel via *mvksdensity*, as a custom kernel, was the slowest implementation (**S3 Fig**). For these reasons we generated KSDE maps using the MATLAB function, *mvksdensity*, and its built-in kernel.

This method depends on three main parameters: the smoothing kernel, smoothing strength, and query points. The smoothing kernel can take many shapes (e.g. uniform, triangular, biweight, triweight, Epanechnikov) but we restricted our analyses to a Gaussian kernel as this was the form proposed by Leutgeb et al. [50,51]. The smoothing strength, often denoted as *h* and called the smoothing 'bandwidth', effectively changes the relative standard deviation of the smoothing kernel, and thus the smoothness of the estimate. Although kernel density estimates are continuous and can be quantified at any point, in order to produce a discretized map they must be queried at a finite set of points. For simplicity, these form a square grid similarly to the bins of a histogram. The density of the query points can easily be controlled by setting the distance between neighboring points, hereafter called the bin size. Unlike with the previous approaches, values are calculated independently for each query point, meaning that decreasing the bin size increases the resolution of the estimate without changing its shape. In this way, we know *a priori* that smaller bin sizes are more accurate and should thus produce the lowest error but, this will likely increase the computation time considerably making small bin sizes sub-optimal. For this reason we tested query grids with bin sizes ranging from 2.5 to 640 mm and bandwidth values ranging from 10 to 320 mm (Methods: *Kernel smoothed density estimate*).

## Temporal KSDE

Brun et al. [56], Fyhn et al. [6] and Leutgeb et al. [57] described a unique spatial mapping procedure which uses a KSDE approach applied to the instantaneous firing rate of the cell, rather than treating all of the data within a bin as a homogenous temporal sample. This temporal KSDE (tKSDE) approach is dependent on three inputs: grid size, spatial smoothing strength, and temporal smoothing strength. The instantaneous firing rate of the cell in time is estimated using a sliding window, typically employing a 2-second-long Blackman window–this approximates a Gaussian with a standard deviation equal to ¼ the window length. However, we found that shorter time windows resulted in more accurate maps overall (**S4 Fig**) and so instead focused on a window duration of 0.125 seconds (Methods: *Temporal KSDE*). The grid size input acts similarly as in the KSDE and provides the resolution at which to estimate the cell's firing rate (i.e. the bin size). The value of each bin is calculated as the weighted average of the cell's instantaneous firing rate, where the values are weighted according to their distance from the bin center, much like in a KSDE. The weighting kernel is typically a second Blackman window, but for simplicity we replaced this with a Gaussian kernel. The spatial smoothing input, σ, defines the standard deviation of this Gaussian and thus acts as a tunable spatial smoothing parameter. For this approach, we used bin sizes ranging from 2.5 to 640 mm and smoothing values ranging from 10 to 640 mm (Methods: *Temporal KSDE*).

## Quantifying rate map error

For each mapping method we generated firing rate maps for our simulated place cells using a range of parameters chosen to encompass values reported in the literature (**Fig 6**). For each rate map parameter combination we then compared the resulting rate map *r(x,y)* to the

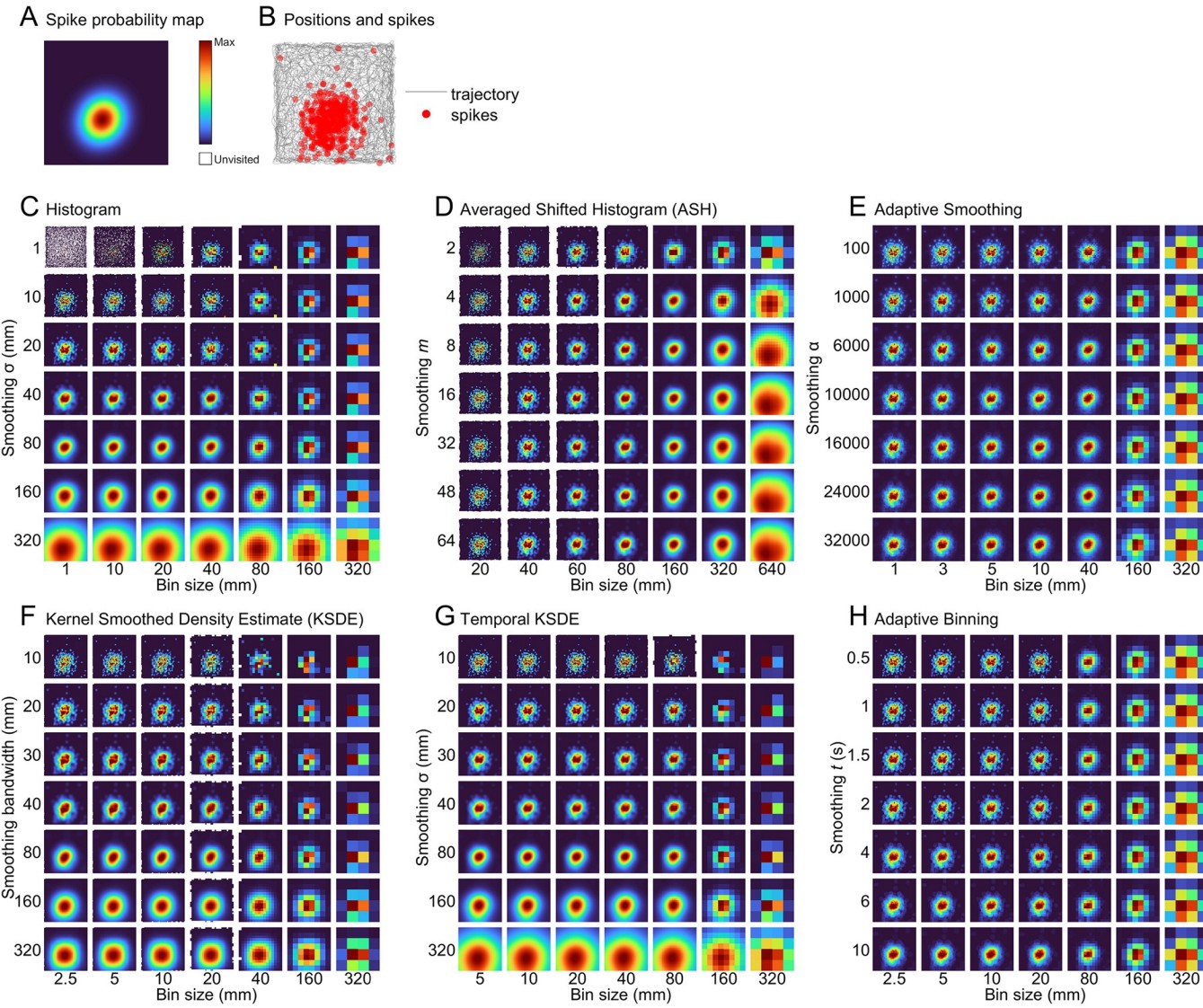

**Fig 6. Qualitative assessment of firing rate maps. A)** The spike probability map of one example simulated place cell. **B)** Spikes simulated in a 16-minute recording session using this probability map. **C-H)** For each mapping method, firing rate maps generated for the spike and position data in b using a range of parameter combinations. For each method, 256 place cells were simulated and mapped in this way.

underlying spike probability of the cell *f(x,y)* using mean integrated squared error (MISE; Methods: *Map accuracy*; note that MISE is in units of integrated probability squared, not firing rate). In testing, similar results were observed when using Pearson correlation, Euclidean distance and mutual information (**S5 Fig**). Every mapping method exhibited some parameter combinations which performed better than others. Generally, small bin sizes coupled with little smoothing, large bins, and high smoothing were all associated with inaccurate maps (**Fig 7A–7F**). It is clear from the example maps in **Fig 6** why this is the case: small bin sizes with little to no smoothing are very sparse, exaggerate small-scale firing features and the lack of smoothing does not allow interpolation between filled bins. Conversely, very large bin sizes or smoothing values provide maps which lack detail and obscure place fields. However, for each method a region of high accuracy maps was apparent between these zones. For the histogram method, smoothed maps were superior in terms of accuracy than when smoothing was

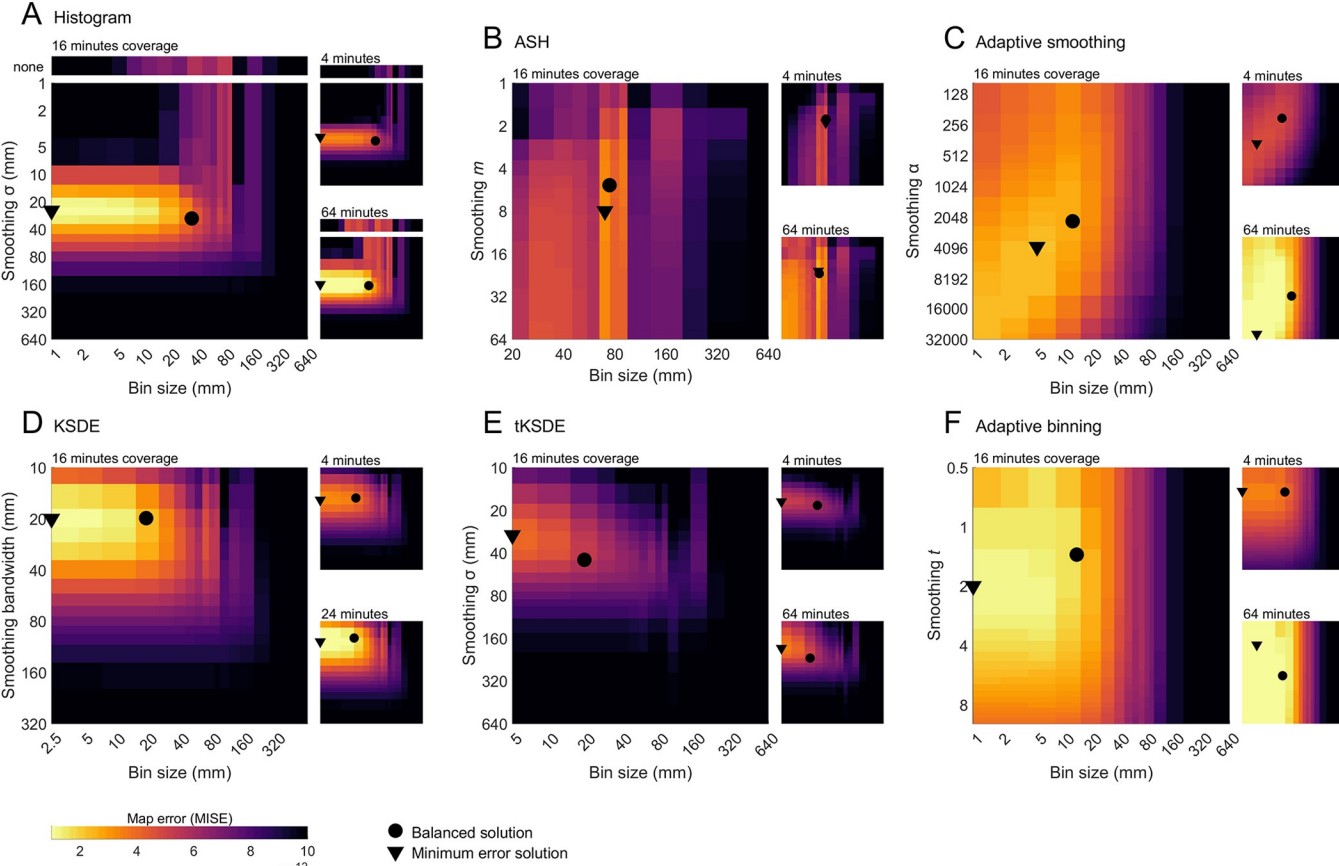

**Fig 7. Quantitative investigation of firing rate map accuracy. A-F)** For each mapping method, the error associated with a range of parameter combinations when used on 16 min duration sessions (main panel), 4- or 64-min-duration sessions (inset right). Each plot is an average of 256 simulated place cells. Also denoted are the parameter combinations associated with the minimum error and the combinations found to optimize error, computation time and place field detection accuracy in a balanced way. Note that plots are shown using a consistent color axis which may not span the full range of data values for every method.

omitted (parameter pair with the smallest average error: with smoothing = $3.1 \times 10^{-12}$, without smoothing = $1.0 \times 10^{-12}$; $t(510) = 14.7$, $p = 1.6 \times 10^{-12}$, two-sample t-test).

Broadly, we found similar results for 4-, 16- and 64-minute-long sessions, although as the session duration was increased the region of low error combinations generally widened to encompass a greater range of parameter combinations (**Fig 7A–7F**). This is to be expected because a firing rate map is estimating $f(x,y)$: the probability a cell will fire at a position given infinite time. Thus, as the number of available samples approaches infinity this estimate can be made with greater certainty. This effect was greatest in the adaptive smoothing and binning methods, which is perhaps unsurprising because these approaches were designed to make the greatest use of the available data. Importantly, as recording duration was increased the region of optimal parameter combinations often expanded to include a broader, overlapping set. This suggests that if an experiment involves sessions of different durations, experimenters should optimize their mapping procedures for the lowest data density as these will also generalize to higher densities. Adaptive binning exhibited the broadest region of low error parameter combinations: essentially, any map with a bin size less than 50 mm and a smoothing criterion greater than 0.5 s exhibited very low errors (**Fig 7F**). This makes adaptive binning much more resistant to incorrect or suboptimal input parameters.

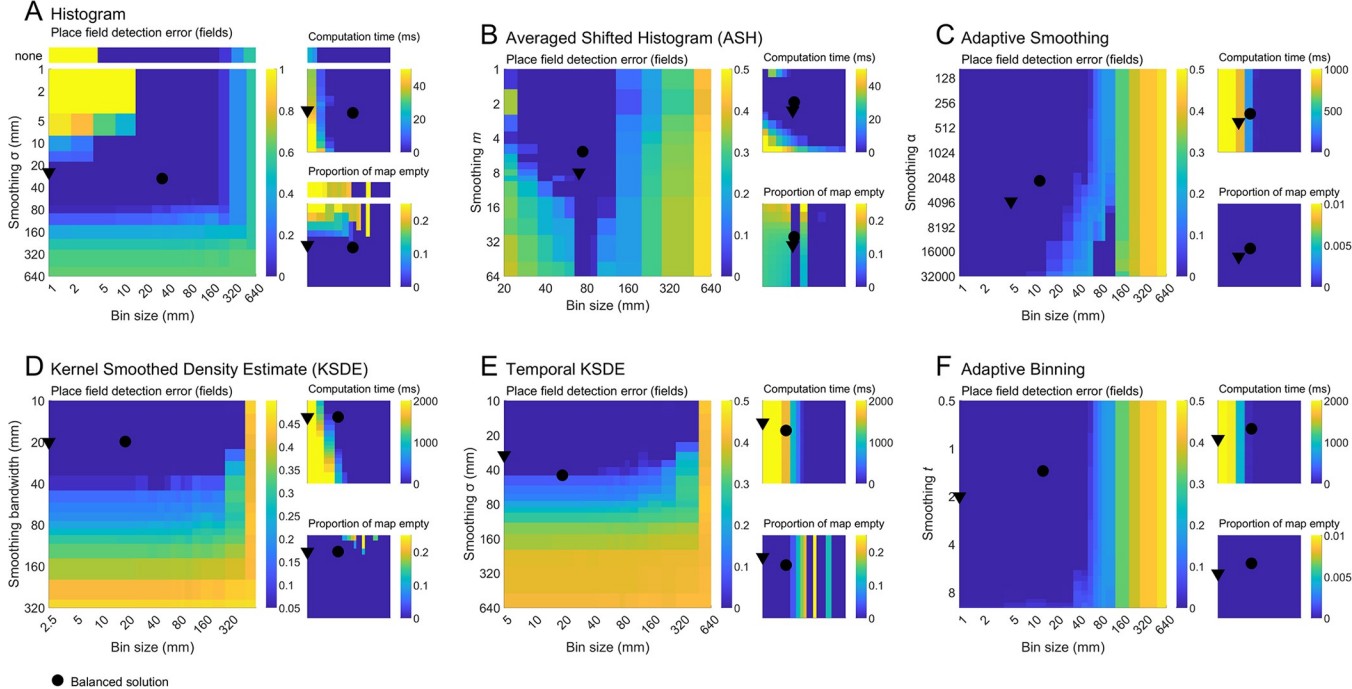

**Fig 8. Additional firing rate map parameters.** All plots are based on the 16-minute session data and are the average of 256 simulated place cells (except temporal KSDE computation time which was estimated using 8 place cells). **A-F)** For each mapping method, the error in detecting place fields (main), the computation time (inset top) and the proportion of the map left empty (inset bottom) associated with each parameter combination are shown. The color axes used here do not show the full range of data values but were chosen to visually isolate low error region(s).

The histogram, ASH and KSDE methods exhibited a region of greater accuracy maps around a bin size of 80 mm (Fig 7A and 7B and 7D), these regions correspond to the bin size at which acceptable accuracy can be obtained with no smoothing (see Fig 6C and 6D) leading to a vertical band of low error maps which is only disrupted by over-smoothing. Interestingly, Sturge's rule [12] suggests that a 16- and 64-minute recording session should be discretized into 70 and 63 mm bins respectively, based solely on the number of position data samples collected (Methods: *Bin size rules-of-thumb*) which closely resembles this 80 mm region. By contrast, two implementations of the Freedman-Diaconis rule [14] suggests that for a 16-minute session, 35- or 18-mm bins should be used and for a 64-minute session 23- and 11-mm bins should be used respectively (Methods: *Bin size rules-of-thumb*). These are close to the optimal bin sizes found for the histogram and KSDE methods (20–30 mm), suggesting that the Freedman-Diaconis rule [14] offers a more accurate rule-of-thumb when generating firing rate maps, although none of these rules-of-thumb are recommended for spatial data.

## Optimizing map parameters

The low error parameter combinations we found sometimes suffered from drawbacks in other areas. For instance, when bin size is changed by a factor of $k$, the total number of bins, and thus computational demand and map size will scale by $(1/k)^2$, thus small bin sizes are typically associated with increased computation time (**Fig 8** insets) and required storage space. These parameters can be ignored when generating a small number of firing rate maps, but they are important to consider when generating maps *en masse*. Generally, error in place field detection was correlated with map error (**Fig 8**; histogram: r = 0.25, ASH: r = 0.78, Adaptive smoothing:

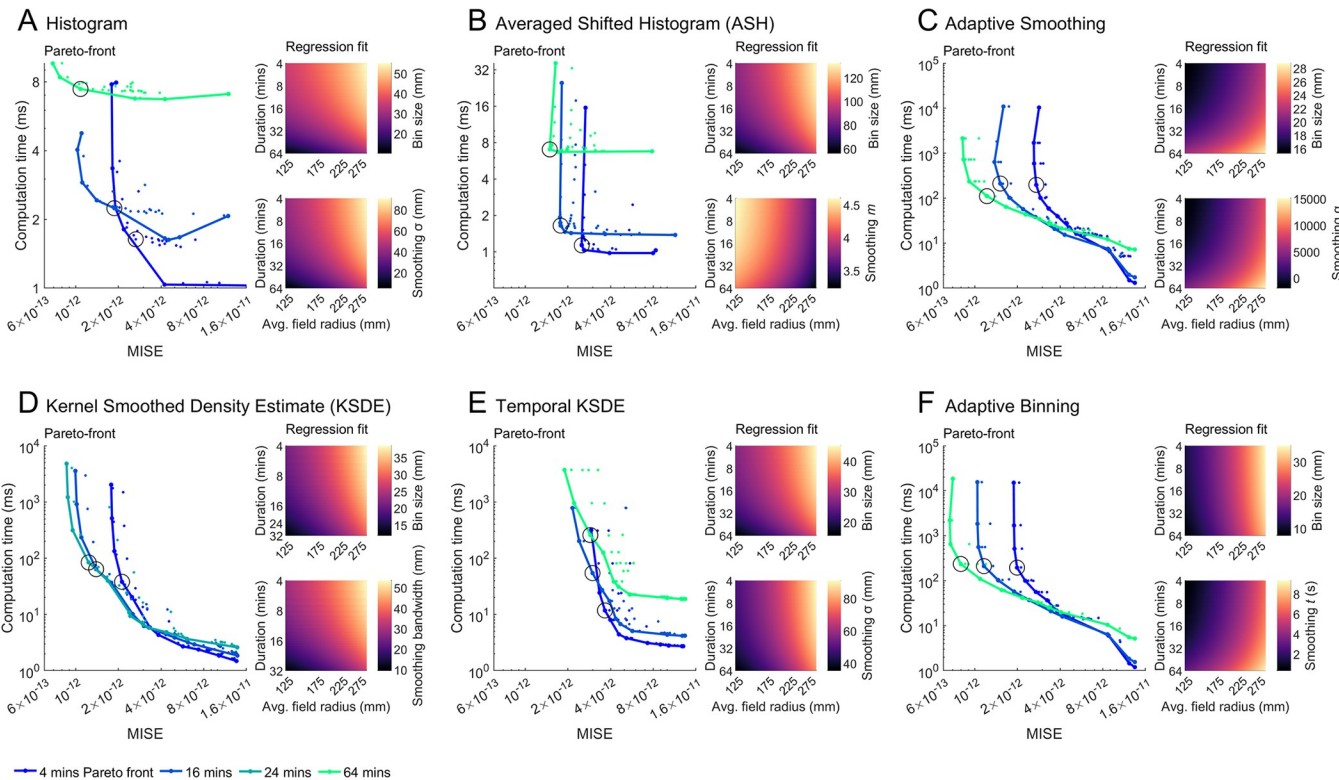

**Fig 9. The relationships between error, computation time and mapping parameters. A-F)** For each mapping method, the Pareto-front of optimal parameter combinations for all tested recording durations. Circled is the parameter combination selected as the most balanced. In addition to computation time and overall map error (MISE), which are plotted here, Pareto-optimization also sought to minimize the proportion of empty bins and place field detection errors. Insets show the relationship between average field radius, recording duration and balanced bin size (top) or smoothing (bottom) parameters, determined using multivariate regression. The exact regression fits can be seen in Table 1 in addition to the fits for the parameters associated with the minimum error.

r = .90, tKSDE: r = 0.54, KSDE: r = 0.72, Adaptive binning: r = 0.98). Small bin sizes were more likely to be associated with a high proportion of empty bins, particularly when little smoothing was used (**Fig 8** insets). Empty bins are not strictly errors and thus do not negatively impact our calculation of map error (MISE), but an abundance of empty bins is often undesirable when investigating a neuron's spatial activity because they represent regions where the cell's activity remains unquantified.

To quantify these complex relationships we used multi-objective (Pareto) optimization [27] to find the combinations of map parameters that not only minimized error but also minimized the computation time taken to generate each map, minimized the proportion of each map that is empty, and maximized the accuracy with which place fields could be detected (Methods: *Parameter optimization*). The Pareto-fronts found for each method can be seen in **Fig 9A–9F**. From among the solutions on this front we selected the one that best balanced error and computation time (hereafter called the 'balanced' solution; Methods: *Pareto-optimal parameters*). In addition, we also found the parameter combination associated with the lowest possible error, ignoring all other factors (hereafter called the 'minimum error' solution). The balanced solutions typically differed from the minimum error solutions by having a much larger bin size but with a similar level of smoothing (**Fig 7A–7F**). This is likely because increasing the bin size while keeping the same level of smoothing generally decreases the computation time while minimally affecting map error and the other performance factors. Maps generated using the two solutions closely matched the underlying spike probability and did not noticeably differ from each other (**Fig 10C**).

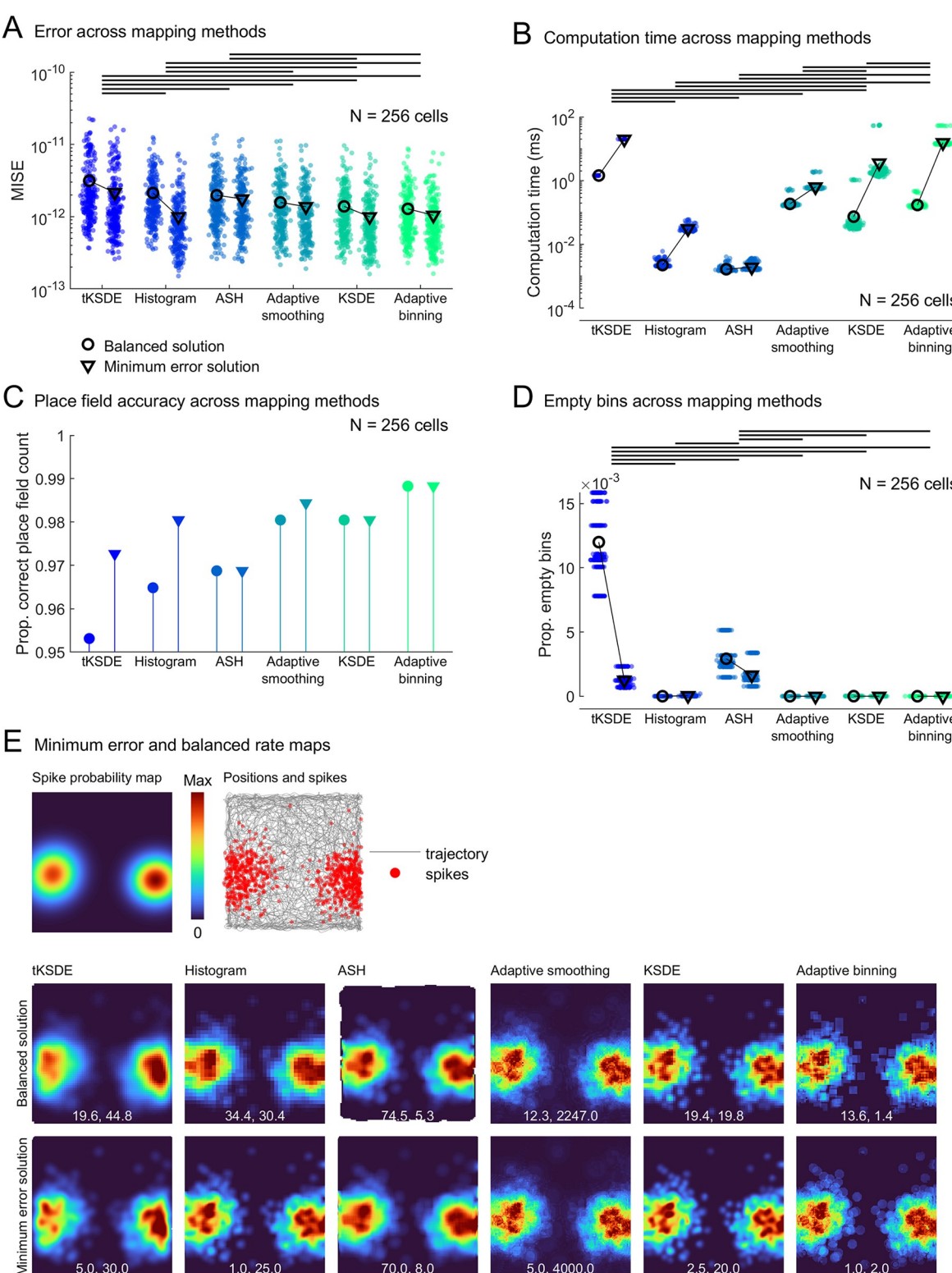

**Fig 10. Error and computation time across methods.** All plots are based on the 16-minute session data. Horizontal lines denote a significant (p < .05) post-hoc comparison, for simplicity only the balanced solution groups are compared. **A)** The mean integrated squared error (MISE) associated with each mapping method when using the balanced or minimum error parameters. Markers denote simulated place cells and the same 256 cells were tested using each method. **B)** The same as a but for computation time. Some methods have a bimodal distribution, the longer computation times represent the first time a map was generated for a recording session (8 unique sessions were simulated) and a dwell time map was created. Only these 8 values are shown for the tKSDE computation times. **C)** The accuracy with

which place fields were detected for each mapping method. **D)** The proportion of empty bins remaining for each mapping method. **E)** The spike probability map and simulated spikes for an example simulated place cell. Below this, firing rate maps for this place cell generated using each method (columns) and the balanced solution parameters (top row) or minimum error solution parameters (bottom row). Text gives the bin size and smoothing parameters used.

## The effects of recording duration and firing field size

We next looked at how recording duration and firing field size affect the selection of mapping parameters. To quantify this relationship, we used multivariate regression (MATLAB *mvregress*); the effect of recording duration and firing field size on the balanced bin size and smoothing strength can be seen in **Fig 9A–9F** (insets). To guide parameter choice more accurately over a wide range of situations, for each method, we have also provided the numerical relationships between recording duration, firing field size, bin size and smoothing strength in Table 1.

Broadly speaking, larger firing fields are best represented by maps produced using larger bins and stronger smoothing (**Fig 9A–9F**, insets), this can also be seen in the positive coefficients of $r$ in Table 1. Large firing fields benefit from large bins and broader smoothing because these emphasize large-scale firing features, such as place fields, rather than small-scale noise. An exception to this is the balanced ASH method, which has a practically flat relationship between field size and smoothing, suggesting that smoothing strength has surprisingly little impact on the resulting map, an effect which can also be seen in **Figs 6D** and **7B**.

Longer recording durations are generally best represented by smaller bin sizes and weaker smoothing (**Fig 9A–9F**, insets), this can also be seen in the negative coefficients of $d$ in Table 1. Low data densities benefit from larger bins and smoothing because this allows better interpolation across sparse regions and increases the accuracy of each bin value by increasing the number of samples their estimate is based on. This is consistent with a general prediction for kernel estimators that as the number of samples increases smoothing should decrease [49,58]. However, the adaptive smoothing and binning methods showed the opposite trend, with longer durations best represented by slightly stronger smoothing and slightly larger bins in the case of adaptive smoothing. Why this is the case is not immediately clear. One possibility is that as the recording duration is increased the region of low error parameter combinations expands markedly for both adaptive methods (Fig 7C and 7F) leading to a more homogenous pattern of error which may have made the regression fit less meaningful for these methods.

**Table 1. The relationship between field size, recording duration, bin size and smoothing strength.**

| Method | Minimum error solution | Balanced solution |
|---|---|---|
| Histogram | *Bin size = smallest available*<br>*Smoothing = 16.232 + 0.119r − 0.218d* | *Bin size = 22.229 + 0.115r − 0.413d*<br>*Smoothing = 9.900 + 0.278r − 0.617d* |
| ASH | *Bin size = 76.339 + 0.055r − 0.278d*<br>*Smoothing = −29.982 + 0.328r − 0.085d* | *Bin size = 56.510 + 0.259r − 0.513d*<br>*Smoothing = 5.338 − 0.006r − 0.006d* |
| Adaptive smoothing | *Bin size = 1.7615 + 0.0077r − 0.0050d*<br>*Smoothing = −11872 + 125r + 281d* | *Bin size = 9.714 + 0.041r + 0.104d*<br>*Smoothing = −9917 + 61r + 106d* |
| Adaptive binning | *Bin size = 1.606 − 0.004r + 0.009d*<br>*Smoothing = −5.078 + 0.041r + 0.050d* | *Bin size = −5.451 + 0.135r − 0.055d*<br>*Smoothing = −4.336 + 0.036r + 0.045d* |
| KSDE | *Bin size = smallest available*<br>*Smoothing = 17.685 + 0.133r − 0.768d* | *Bin size = 9.900 + 0.099r − 0.325d*<br>*Smoothing = 14.806 + 0.139r − 0.707d* |
| Temporal KSDE | *Bin size = 3.529 + 0.014r − 0.020d*<br>*Smoothing = 10.949 + 0.213r − 0.179d* | *Bin size = 14.720 + 0.104r − 0.188d*<br>*Smoothing = 19.304 + 0.242r − 0.222d* |

Table showing regression fit equations for determining the bin size and smoothing associated with the minimum error and balanced solutions given the average firing field radius $r$ and the recording session duration $d$.

In the case of the histogram and KSDE, bin size is absent from the minimum error formula because it was always equal to the smallest tested (Table 1). This is perhaps unsurprising because a histogram generated with the smallest possible bins (i.e. matching the resolution of the position and spike data) and large enough smoothing will approach the result of a KSDE, which is in turn generally considered one of the most accurate density estimation approaches [48,49]. In practice, bin size is limited by the resolution of the camera used to track the animal and the distance of this camera from the experimental apparatus. So, this value can simply be replaced with the smallest bin size permitted by the data or desired by the experimenter. This is permissible because the minimum-error and balanced solutions share practically identical smoothing values and so the resulting map will always fall between these points and approximate the best possible result.

## Estimating field size

The equations presented in Table 1 call for an estimate of the cell's firing field radius, *r*, however, we typically estimate firing field size using a firing rate map. How can we estimate this value *before* we have generated a firing rate map? Firstly, using firing field size values estimated in previous experiments, or using a qualitatively accurate map, will often be sufficient for deriving this value as we found that place field detection was typically resistant to errors in the firing rate map (although the two are correlated). However, it is also possible to estimate the average field radius without a rate map. One way is to use spatial density metrics such as Ripley's k-function (Methods: *Estimating field size*; [59]). This algorithm estimates the radius of dense clusters in *xy* point data, such as the positions of a cell's spikes [60]. Using this method on our simulated place cell dataset, we found a high positive correlation between the actual and estimated place field sizes (**S6B Fig**; r = 0.71, $p = 1.3 \times 10^{-50}$, Spearman's correlation between estimated and real field size). Further research could look to improve on these kinds of density metrics for place field size estimation.

## Comparing mapping methods

We compared several different mapping approaches: the standard bivariate histogram (coupled with Gaussian smoothing), averaged shifted histogram (ASH), adaptive smoothing, adaptive binning, kernel smoothed density estimate (KSDE) and a temporal implementation of the KSDE (tKSDE). Concentrating on the balanced solution results and looking at the error associated with each mapping approach, a one-way ANOVA confirmed that the approaches differed (F(5,1530) = 29.0, $p < .0001$, $\eta^2 = 0.05$), the results of post-hoc multiple comparisons can be seen in (**Fig 10A**). A one-way ANOVA also confirmed that the approaches differed in their computation time (F(5,1284) = 712.8, $p < .0001$, $\eta^2 = 0.74$), the results of post-hoc multiple comparisons can be seen in (**Fig 10B**). A one-way ANOVA also confirmed that the approaches differed in the proportion of bins left empty (F(5,1530) = 4352.6, $p < .0001$, $\eta^2 = 0.93$), the results of post-hoc multiple comparisons can be seen in (**Fig 10D**).

If using balanced parameters, the adaptive binning, adaptive smoothing and KSDE methods demonstrated the smallest average map error (**Fig 10A**). The maps generated using these methods were also the most reliable when detecting place fields (**Fig 10C**), contained very few empty bins (**Fig 10D**), and exhibited a high level of detail (**Fig 10E**), which may make them desirable for investigating fine-scale firing characteristics. However, these methods were also among the slowest to produce (**Fig 10B**) making them undesirable when generating maps *en masse* (the KSDE was so slow we could not test recording durations greater that 24 minutes or bin sizes smaller than 2.5 mm because the computation times became untenable).

The tKSDE, histogram and ASH methods demonstrated the largest average error. However, the histogram and ASH methods were virtually instantaneous to produce, making them a very practical mapping method (**Fig 10B**). When using parameters that minimize error only, the histogram and KSDE methods showed the greatest possible levels of accuracy (**Fig 10A**), in this state the histogram method remains computationally fast while the KSDE method remains one of the slowest (**Fig 10B**). Given that the histogram is the most widely used approach, is very flexible, computationally very fast and provides very accurate maps, it is likely to be best suited to most scenarios. Additionally, the computational speed of histograms makes them very well suited to shuffle-based analyses that are becoming more common in the field [24]. It should be noted however, that the ASH method shares many of these properties and produces much higher resolution maps.

In addition to exhibiting the largest error, the tKSDE approach was also very slow, unreliable for place field detection and among the worst at compensating for poor sampling, making it one of the least desirable methods overall. This approach utilizes the instantaneous firing rate of the cell over time to estimate the average firing rate within a bin making it unique in this aspect, as all the other methods treat data points within or around a bin as if they were contemporaneous. One benefit of this approach is that it relies on an instantaneous firing rate vector which could prove valuable for mapping the results of calcium imaging experiments. In this case, exact spike times are unknown, and researchers use algorithms to estimate them before constructing firing rate maps, but a temporal KSDE would eliminate the need for this step.

## Overdispersion and uneven sampling

These results did not vary greatly when cells were simulated to include heavy overdispersion [61,62], although the balanced and minimum error solutions consistently shifted towards significantly stronger smoothing (**S7 Fig**). This increased smoothing requirement likely reflects the larger uncertainty associated with overdispersion and should be taken into account when analyzing recording sessions of a short duration and when overdispersion is anticipated. Uneven trajectories, such as those caused by goal-directed navigation or thigmotaxis, did not change the results of the bin size and smoothing analyses (**S8 Fig**) although MISE did increase significantly when trajectories showed strong goal-directed biases. Lastly, to test the robustness of our results in the face of real-world place cell data, we used the equations provided in Table 1 to calculate firing rate map parameters for real place cells extracted from the dataset of [20] (**S9 Fig**).

## Comparison to the literature

Having now quantified the error associated with different input combinations, we looked back at the parameter combinations reported in the spatial navigation literature. To do this we surveyed peer-reviewed papers that focus on spatial neurons, spanning from the first description of a firing rate map (1983) to the current year (2023). We recorded their rate map approach and the parameters used (Methods: *Literature survey*).

Focusing first on the histogram method, and looking at the types of smoothing used, we found that over time researchers have gradually transitioned from no smoothing to boxcar kernels and finally to Gaussian kernels, suggesting that the field has recognized the increased accuracy afforded by smoothing (**Fig 11A**). We also found a weak, but significant, decrease in bin sizes over time, likely related to the adoption of more powerful laboratory computers and better smoothing techniques (r = -0.2, $p$ = .016; **Fig 11A**, black line). We also found that when Gaussian smoothing was used, the parameter combinations reported in the literature closely

 

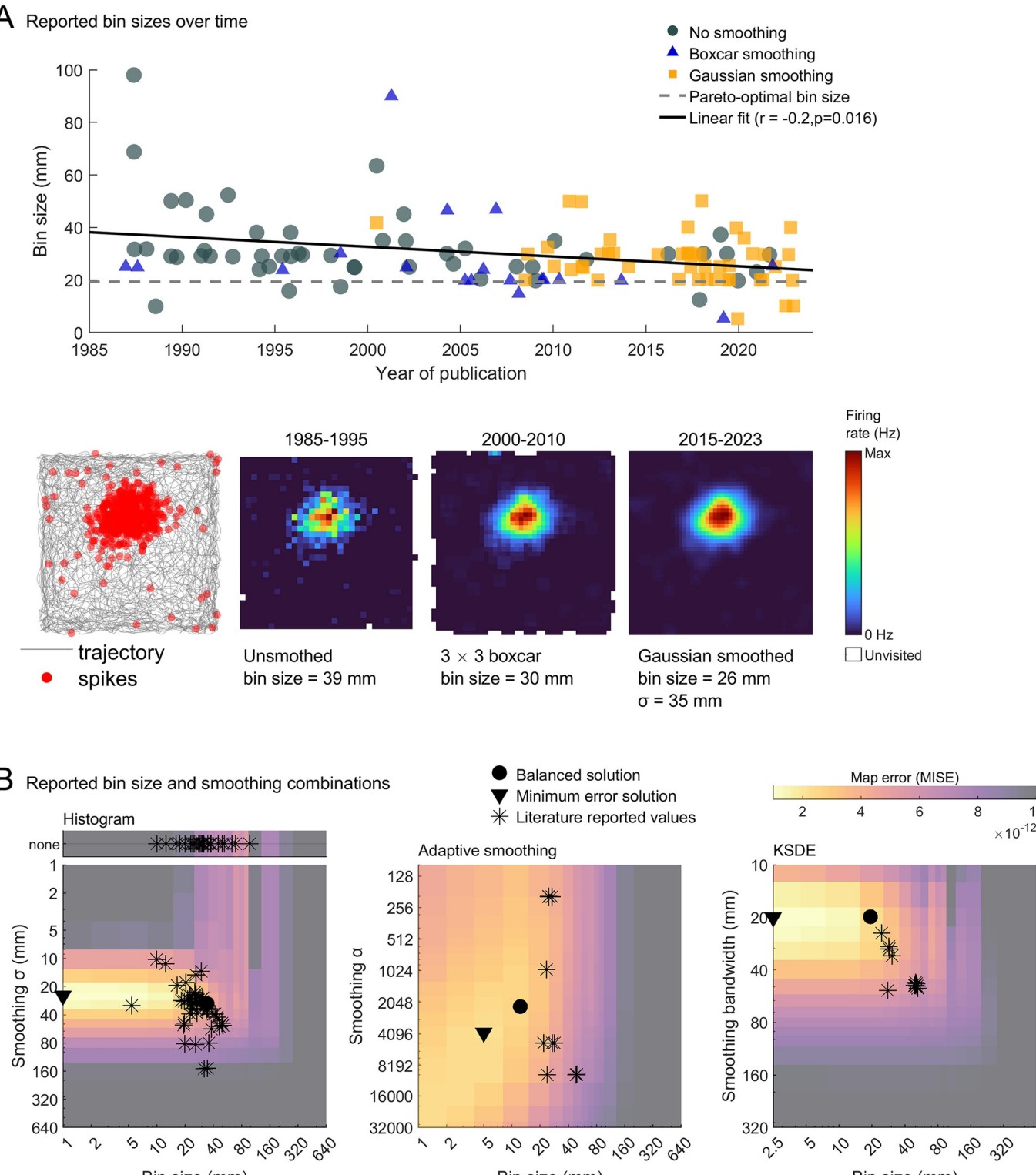

**Fig 11. Mapping parameters in the published literature.** A) Top: bin sizes used to generate histogram firing rate maps in the literature, surveyed across 100 published papers and plotted as a function of publication year. Gaussian jitter (mean = 0, σ = 2) was added to the literature values to make visualization of overlapping data clearer. Bottom: the spike and position data for an example, simulated place cell. Firing rate maps for this cell are depicted to the right, generated using the average parameters used in the specified time period. B) Error maps overlaid with the minimum error, balanced and literature reported parameter combinations for the histogram, adaptive smoothing and KSDE methods as in **Fig 7**. In the published literature, researchers tend to use parameter combinations

which are very close to the balanced solution. Gaussian jitter (mean = 0, σ = 2) was added to the literature values to make visualization of overlapping data clearer. See **S10 Fig** for example ratemaps generated using different literature values.

clustered around the balanced solution found through our analysis (**Fig 11B**) although they still showed a wide range of variability between studies (**S10 Fig**). When smoothing was not used the reported bin sizes still tended to cluster in the lowest error region (**Fig 11B** inset top). This suggests that researchers have, generally, been remarkably successful at balancing rate map error with computation cost. Place field detection was also very accurate across the parameter combinations reported in the literature (**S11 Fig**), this may be because place field detection represents only a very rough measure of map accuracy, or because researchers using firing rate maps optimize their firing rate maps for accurate place field detection rather than computation time or MISE.

Adaptive smoothing and KSDE are the next most widely adopted mapping methods and show a similar clustering around the balanced solution (**Fig 11B**). However, with these methods researchers tend to use over-sized bins, likely to offset the low speed of these approaches.

## Summary

Here we replicated the most widely adopted firing rate map methods described in the literature, as well as some rarely and never-before used ones, and quantified the effects of their different input parameters. We did this by generating maps for simulated data sets and comparing the results to their ground-truth firing probability distribution. For each method we determined the parameter set yielding the smallest mean integrated squared error possible, then we considered factors such as computation time and place field detection error to determine the Pareto-optimal parameter set: settings which balanced error and computation time.

Each approach exhibited a clear region of optimal parameters. Typically, the solution associated with the lowest error consisted of the smallest bin size possible and a relatively large amount of smoothing; in comparison the balanced solutions tended to call for a larger bin size with less smoothing. Overall, we found that the bivariate histogram method was consistently among the fastest while still providing surprisingly accurate maps. The ASH method was consistently the fastest and provided a medium level of accuracy, coupled with increased spatial resolution, these results suggest that the ASH method may be an unfairly overlooked option for generating rate maps. Adaptive smoothing and binning approaches compensated for low sampling the most effectively, as expected, and produced lower error maps but at a longer computation time. The histogram and KSDE approaches achieved the greatest possible accuracy when all other factors were ignored. However, the extremely long computation time of KSDE makes it a generally undesirable choice.

## Limitations and further research

Here we have concentrated on widely used approaches for generating firing rate maps, with some exploration of underused variants. However, there are a number of more recent and more advanced approaches which have not yet received wide adoption. These include Bayesian [63] and Gaussian process [64,65] methods. These computationally complex approaches could provide improvements over traditional mapping methods and will require further investigation. Increasing the adoptability of these approaches would facilitate this; for example, we have provided MATLAB code alongside this paper which allows the reader to adopt any of the methods discussed here with minimal effort and without a need for any computational background. Complex approaches often provide a mathematical description but lack a packaged implementation, which likely hinders their adoption by the research community.

In this analysis we have concentrated on an open field arena setup which represents one of the simplest and most widely used experimental apparatus in the literature [16,19,22,66,67]. While we do not expect the findings here to differ among open field environments of a different shape, many experiments instead make use of complex alleyway mazes [68–70] where place fields can take on new characteristics [71,72]. These mazes also present new challenges for spatial mapping, such as areas separated by walls [73,74] that should not be smoothed together. While most experiments solve this problem by linearizing the maze data, this is not always possible or desirable. For a more flexible solution these situations would benefit from non-isotropic smoothing kernels such as the alleyway-limited smoothing employed by Derdikman et al. [68] or geodesic kernels that take into account the shape of an environment during smoothing. Future research will be needed to develop these alleyway-suited mapping methods.

## Conclusion

While the purpose of a firing rate map is to quantify the spatial activity of a neuron, it is surprising that these maps are often based on qualitatively chosen parameters. Here we have shown that, while generally accepted, this should not be the case because some parameter combinations are clearly superior. In the field, it is difficult to know which parameters to choose and often a multitude of such choices must be made across many different analyses. We have provided a way to guide at least some of these in the future. Additionally, we hope that this investigation will motivate further inspection and discussion of these, and perhaps other tools which the spatial navigation field relies on.

## Methods

### Random walk

To simulate rat exploratory behavior, we employed a Gaussian random walk. The environment was recreated as a binary grid map, with a resolution of 0.5 cm. The agent started in the geometric center of the environment with a random starting heading, it then made successive steps through the environment, with each step representing 1 second of walk time for a total duration of 64 minutes. At each time point the agent's next position was calculated as:

$$g(x) = argmax\left(\frac{1}{\sigma\sqrt{2\pi}}e^{\frac{-(D-\mu)^2}{2\sigma^2}} + \frac{1}{2\pi I_0(\kappa)}e^{\kappa\,\cos(A-a_{t-1})} + occ + d_w + d_c + b\right) \quad (1)$$

where $D$ is the distance to every pixel in the grid map from the agent's current position, $\mu = 64$ and $\sigma = 128$, $I_0$ is the modified Bessel function of order zero, $A$ is the angle to every pixel in the grid map from the agent's current position, $\alpha_t$ is the angular heading of the agent at time step $t$, $\kappa$ is the concentration of the von-Mises distribution which was set to 1, $occ$ is the total occupancy in each grid map pixel calculated using a retrospective 8-minute sliding window, $d_w$ and $d_c$ are the distance to every pixel from the nearest wall in the environment and the distance to every pixel from the center of the environment respectively, $b$ was an optional parameter used in the biased sampling conditions and is described below.

Each input described above was normalized to range from zero to one (the maximum value). Additionally, the values of $occ$ were inverted. $d_w$ and $d_c$ were Gaussian weighted according to the function:

$$f(x|\mu, \sigma) = \frac{1}{\sigma\sqrt{2\pi}}e^{\frac{-(x-\mu)^2}{2\sigma^2}} \quad (2)$$

For $d_c$ $\mu$ was set to 0 and $\sigma$ was set to 512, for $d_w$ $\mu$ was set to 0 cm and $\sigma$ was set to 512.

To simulate a goal-directed trajectory, biased towards one position in the environment, *b* was included and set to the distance to every pixel from a single point in the environment located halfway between the minimum and maximum y-value and three quarters of the way between the minimum and maximum x-value. *b* was also Gaussian weighted using the function in (2), *μ* was set to 0 and *σ* was set to 25, the result was also down weighted by a factor of 0.9. To simulate thigmotaxis, $d_c$ was set to zero and for $d_w$ *σ* was set to 100 so that the agent would be attracted towards the walls more frequently.

Once *g(x)* was found, the quasi-Euclidean optimal route between the current and next point, avoiding obstacles, was taken as the agent's path and this was up sampled by a factor of 50 (for a 50Hz positional sampling rate). This step allows the agent to navigate around barriers and obstacles efficiently and smoothly, although this was not needed in the current analysis. Lastly, to mimic errors in real-world position tracking we added random jitter between ±8.5cm to the points and smoothed the resulting path using an unsupervised, robust, discretized, n-dimensional spline smoothing algorithm (MATLAB function *smoothn* [75,76] with the smoothing parameter set to 128).

In effect, at every time step the agent 'chooses' a next step which does not deviate too much from its current heading and is not too far for a rat to travel in the given time. It is also weakly attracted towards the center of the environment, weakly repelled by walls in the environment and attracted towards any areas of the environment not visited recently. To represent the fact that place cell datasets are recorded across multiple animals and sessions, we simulated 8 different random walks using the settings above. To investigate the effect of sampling we tested 4-, 16- and 64-minute-long sessions, these were generated by simulating a 64-minute long session as described above and then clipping the sampling to the desired duration from the start of the session.

## Place cells

Place cell spike probability distributions were generated as the sum of one or more multivariate normal distributions (MATLAB *mvnpdf*). Three field sizes were simulated to reflect the fact that the diameter of place fields varies along the dorso-ventral axis of the hippocampus [77] and between brain regions [1]. These groups had a mean standard deviation in the x- and y-dimensions of 4000, 8000 and 1600 respectively, or in other words an average field radius at 2 standard deviations of 126, 179 and 253 mm. To reflect individual variability among cells these values were drawn from a normal distribution (MATLAB *normrnd*) with a standard deviation of 1000 and limited to values greater than 1000. Additionally, to reflect the fact that spatial fields are not always isotropic, but are often elongated [19,20], standard deviation in the x- and y-axis were determined independently. To emphasize this elongation, but also allow fields to adopt orientations oblique to the x- and y-axes fields were also assigned an *xy* covariance. This was calculated as the average variance in *x* and *y*, multiplied by a value drawn from a normal distribution (MATLAB *normrnd*) with a mean of 0, a standard deviation of 0.25 and limited to values between –1 and 1. Field centroids were randomly distributed within the simulated environment.

For each place cell a spike probability map was modelled as the random combination of these multivariate normal distributions, 512 of which were generated in total. The number of fields expressed by each cell was drawn from a gamma distribution (MATLAB *gamrnd*) with shape parameter 5.73 and a scale parameter 0.26, limited to values greater than 1. These fields were then randomly selected from the pool of 512 and merged into one spike probability map. Merging was done by taking the maximum value across the spike probability maps at each location. In this way we simulated 256 place cells for each field radius group.

## Spiking

Each place cell was randomly assigned to one of the 8 random walk trajectories we described above (Methods: *Random walk*). Next, to model spiking, the value of its spike probability map (Methods: *Place cells*) was found for each position in this random walk. This value was then converted to a spike count by drawing a value from the Poisson distribution (MATLAB poissrnd) with λ set to the spike probability value. Additionally, to mimic realistic single-unit recordings we combined this value with a random background noise value also drawn from the Poisson distribution (MATLAB poissrnd) with λ set to 0.001. To mimic errors in position tracking and the local drift of place cell activity within a theta cycle [78] we also randomly time-shifted spike probability values, forwards or backwards along the animal's trajectory, by a duration which was drawn from a normal distribution (MATLAB *normrnd*) with a mean and standard deviation of 0.01 and 0.02 seconds respectively.

Lastly, the spikes resulting from this procedure were downsampled to match each cell's desired mean firing rate. For each cell the desired average firing rate was drawn from a normal distribution (MATLAB *normrnd*) with a mean of 1, a standard deviation of 1 and these were limited to values greater than 0.5 but less than 10 Hz.

To simulate overdispersion [61,62] the above steps remained the same but we employed a method similar to that reported by [62]. Before generating a spike train, the spike probability vector was multiplied by a 1.5 Hz time-varying signal which varied between 0.2 and 1.8 (either decreasing or increasing the likelihood of spikes occurring by 80%).

## Histogram

Formally, where $x_0, y_0$ is the origin of the histogram and $h$ is the side length of a square bin, the $ij$th bin of the histogram is defined as the left-closed right-open interval:

$$B_{ij} = [x_0 + ih, x_0 + (i+1)h) \& [y_0 + jh, y_0 + (j+1)h) \tag{3}$$

if the number of spikes falling into $B_{ij}$ is denoted by $S_{ij}$ and the number of position samples as $P_{ij}$ then a firing rate map would be defined as:

$$r(x,y) = \frac{S_{ij}}{P_{ij}(f_s)} \; for \; x,y \in B_{ij} \tag{4}$$

where $f_s$ is the sampling interval of the position data. For smoothing we implemented a Gaussian smoothing kernel defined as:

$$g(x,y) = e^{-\frac{x^2+y^2}{2\sigma^2}} \tag{5}$$

where $x$ and $y$ denote the bin position relative to the one being smoothed and $\sigma$ is the standard deviation of the distribution. Throughout our analyses the kernel size was always set to:

$$kernel\ width = 2\lceil(2\sigma)\rceil + 1 \tag{6}$$

ensuring that the kernel was always an odd number of pixels and at least 4 standard deviations in width. Maps were padded with zero values before smoothing to accommodate the smoothing kernel and reduce edge effects. We chose zero-padding because it most closely represents the actual data values collected without attempting to extrapolate spatial activity outside the sampling range. However, other padding methods such as reflective or symmetric padding may be preferred, especially when the out-of-field firing rate of a cell deviates significantly from zero, and these can be implemented using the code provided alongside this paper. We chose a Gaussian smoothing process as it should approximate the Gaussian shape of place

fields well [79], it is increasing in popularity in experimental studies (**Fig 11A**) and it allows for the most straightforward comparison with the other rate mapping methods described later. We provide MATLAB code alongside this paper for generating histogram maps described in the literature but with a number of time saving modifications (*rate_mapper*; 'histogram' method; Fig 2).

As a baseline we also tested histograms without any smoothing applied. Additionally, we also tested the result of applying smoothing on the firing rate map, rather than the spike and dwell maps. For this latter approach, a firing rate map was calculated using unsmoothed spike and dwell maps, and then the Gaussian function described above was applied (**S1 Fig**). Smoothing after the division of spike and time values is complicated by the inclusion of missing values–unvisited bins result in a division by zero and are typically considered unvisited and filled with placeholder values such as *NaN*. Ordinarily, convolution of a kernel with a map containing *NaN* values results in propagation of these values, resulting in a degraded map. Thus, for smoothing after the division we made use of the MATLAB function *nanconv* [31] which ignores *NaN* values during convolution. This variant of the histogram approach can also be replicated using the supplied MATLAB code.

## Averaged shifted histogram (ASH)

Formally, to compute the ASH, a grid of square bins is defined as for the bivariate histogram with side length $h$. Each bin is then further subdivided into $m$ smaller sub bins each having a side length of $\delta = h/m$. If the $ij$th sub bin is defined as:

$$B_{ij} \equiv [i\delta, (i+1)\delta)\&[j\delta, (j+1)\delta) \tag{7}$$

and *nij* is the number of samples falling within this bin, then the bivariate ASH is defined as:

$$\hat{f}(x, y) = \frac{1}{nh^2} \sum_{i=1-m_x}^{m_x-1} \sum_{j=1-m_y}^{m_y-1} w_{m_x}(i) w_{m_y}(j) v_{k+i,l+j} \tag{8}$$

where $w$ is the weighting function:

$$w(i) = \frac{m \cdot k(i/m)}{\sum_{j=1-m}^{m-1} k(j/m)} \tag{9}$$

in our case $k$ is the quartic or biweight kernel, although other kernels can easily be employed instead:

$$k(t) = \frac{15}{16}(1 - t^2)^2 \tag{10}$$

In effect, a kernel is passed over every sub bin which calculates the weighted average of the local sub bins within a region of side length $h$ (the original bin size). As long as the kernel satisfies the conditions of symmetry, nonnegativity and its weights sum to 1 the resulting histogram will still sum to the number of input samples [80]. A different smoothing kernel function $k(t)$ can also be applied [81] but we used the quartic or biweight kernel as it efficiently approximates a Gaussian and is the kernel suggested by [38]. We padded maps with zero values before applying the kernel to reduce edge effects. We chose zero-padding because it most closely represents the actual data values collected without attempting to extrapolate spatial activity outside the sampling range. However, other padding methods such as reflective or symmetric padding may be preferred, especially when the out-of-field firing rate of a cell deviates significantly from zero, and these can be implemented using the code provided alongside this paper. We provide MATLAB code alongside this paper for generating ASH maps (*rate_mapper*; 'ash' method).

## Adaptive smoothing

Formally, in this approach each bin is incrementally expanded outwards and at each increment the number of position data and spikes contained within the circle are counted. This continues until the following formula is satisfied:

$$r \geq \frac{\alpha}{n_p \sqrt{n_s}} \tag{11}$$

where $r$ is the radius of the circle, $\alpha$ is a parameter which controls the smoothing extent of the bins, $n_p$ is the number of position data points falling within the circle and $n_s$ is the number of spikes falling within the circle. Once a radius has been found which satisfies the adaptive equation the firing rate of the bin is calculated using the spikes and position samples within that radius. After iterating across all bins, the result is a firing rate map with a regularly spaced grid of values, based on differently sized regions of underlying data which may also be overlapping. Note that the version reported by Skaggs et al. [43] took the form:

$$n_s > \frac{\alpha}{n_p^2 r^2} \tag{12}$$

This is still often cited but requires orders of magnitude larger values for $\alpha$ (i.e., $1 \times 10^{13}$ in one study). Note that both methods will fail without intervention if $n_s$ is zero–some studies, such as Yoganarasimha et al. [82] exclude cells with a low number of spikes.

The computational resources needed to generate adaptive rate maps can be far larger than a bivariate histogram. For each bin the distance to all position and spike data points needs to be known so that the number of points falling within a radius $r$ can be calculated. For large datasets this distance calculation can be computationally expensive. Furthermore, the resolution at which $r$ is tested is unclear–by what amount should the bin expand with each iteration? We provide MATLAB code alongside this paper for generating adaptive smoothed maps as described in the literature but with a number of time saving modifications (*rate_mapper*; 'adaptive' method; **Fig 3**).

However, we used a convolution-based implementation of the adaptive smoothing method which takes a fraction of the computation time with little to no change in the resulting firing rate maps (*rate_mapper*; 'kadaptive' method; Figs **4** and **S2**). This implementation first bins the position and spike data using the unsmoothed histogram method. The $n_p$ and $n_s$ values of the adaptive equation are then calculated by convolving the histogram with a circular kernel of radius $r$, which sums the values of the bins within its radius. This process is then repeated for kernels with varying values of $r$. We used 32 kernel radii varying from the width of a single histogram bin to 640 mm. The smallest kernel radius which satisfies the adaptive equation can then be used to calculate the firing rate of the cell in that bin. To reduce edge effects, in our implementation we pad maps with zero values before applying the kernel. We chose zero-padding because it most closely represents the actual data values collected without attempting to extrapolate spatial activity outside the sampling range. However, other padding methods such as reflective or symmetric padding may be preferred, especially when the out-of-field firing rate of a cell deviates significantly from zero, and these can be implemented using the code provided alongside this paper.

## Adaptive binning

Adaptive binning, described by Yartsev and Ulanovsky [44] and Ginosar et al. [45], is very similar to adaptive smoothing. As with adaptive smoothing, for each bin a circle is incrementally expanded outwards, but in this case at each increment only the number of position data

contained within the circle are counted. This continues until the bin contains equal to or more than *t* seconds of data, where *t* in this case is a tunable smoothing parameter. Once a radius has been found which satisfies this constraint, the firing rate of the bin is calculated as the ratio of spikes and position samples within that radius. After iterating across all bins, the result is a firing rate map with a regularly spaced grid of values, based on differently sized regions of underlying data which may also be overlapping.

As before, we provide MATLAB code alongside this paper for generating adaptive binned maps described in the literature but with a number of time saving modifications (*rate_mapper*; 'yadaptive' method). However, we used a convolution-based implementation of the adaptive binning method which takes a fraction of the computation time with little to no change in the resulting firing rate maps (*rate_mapper*; 'kyadaptive' method; similar procedure to Figs 4 and S2). This implementation first bins the position and spike data using the unsmoothed histogram method. The number of spikes and position data falling within each radius of a bin are then calculated by convolving the histograms with a circular kernel of radius *r*, which sums across the bins falling within it. This process is then repeated for kernels with varying values of *r*—we used 32 kernels varying from the width of a single histogram bin to 640 mm in width. The smallest kernel width which contains *t* seconds of data can then be used to calculate the firing rate of the cell in that bin. To reduce edge effects, in our implementation we pad maps with zero values before applying the kernel.

One disadvantage of adaptive smoothing and adaptive binning is that they do not natively produce a spike or dwell time map. This can be an issue if researchers want to calculate spatial measures such as spatial information content [43] which depends on comparing a rate and dwell map. It can also be an issue if researchers need a dwell time map to quantify coverage or biases in sampling position. It is likely in these cases that a separate set of dwell maps will need to be produced based on another method, such as the bivariate histogram.

## Kernel smoothed density estimate

Formally, in this approach the firing rate map is no longer made up of 'bins' but rather it is estimated at a grid of equally spaced points overlaid on the data, but for simplicity we will refer to these grid points as bins. For each bin the probability density of a dataset *x* is estimated as:

$$\hat{f}(x) = \frac{1}{nh} \sum_{i=1}^{n} K\left(\frac{x - y_i}{h}\right) \tag{13}$$

where $y_i$ are the bin centers, *n* is the total number of data points, *h* is a tunable smoothing parameter (also known as the 'bandwidth', 'window width' or 'smoothing parameter' [40]) and *K* is the kernel density function which can be any symmetric function satisfying:

$$\int_{-\infty}^{\infty} K(x)dx = 1 \tag{14}$$

and is often a Gaussian kernel.

There is no implementation of the KSDE designed to deal with two separate but concomitant data sets, instead Leutgeb et al. [50,51] used this approach to generate a firing rate map as the ratio of a KSDE calculated on spike data (i.e. a spike map) and a KSDE calculated on position data (i.e. a dwell time map) separately:

$$\hat{f}(x) = \sum_{i=1}^{n} K\left(\frac{s_i - x}{h}\right) \bigg/ \int_{0}^{T} K\left(\frac{y(t) - x}{h}\right) dt \tag{15}$$

where, as before, *K* is a smoothing kernel, *h* is the smoothing parameter, *n* is the total number

of spikes, $s_i$ is the location of the i-th spike, $y(t)$ is the location of the rat at time $t$, and [0 T] is the entire recording period. Although the numerator calls for integrating across the position probability density, in practice this is also evaluated numerically (e.g. through the trapezoidal method as a sum). Unconstrained, this approach will produce a firing rate map with no empty bins, as both probability densities may approach but will never equal zero. In practice, bins that are more than a set distance away from any position data (i.e., 50 mm) are typically set as empty.

The computational resources needed to generate KSDEs can be quite large. For each bin the distance to all position and spike data points needs to be known and then Gaussian weighted. For large datasets this calculation can be computationally expensive. We provide MATLAB code alongside this paper for generating KSDE maps using the Leutgeb et al. [50,51] approach and a native (MATLAB function *mvksdensity*) approach, with a number of time saving modifications (*rate_mapper*; 'leutgeb', 'leutgeb_pixelwise' and 'ksde' methods respectively; **Fig 5**). The *mvksdensity* implementations both utilize the 'reflection' boundary correction method to reduce edge effects. We employ a Gaussian smoothing kernel as this is the form described by [50,51].

Although not investigated here, a similar approach is described by Harris et al. [83] which can be formalized as:

$$f(x) = \frac{\sum_{i=1}^{n} n_t K(x_i - x)}{\sum_{i=1}^{n} K(x_i - x)dt} \tag{16}$$

Here, $K$ is a weighting kernel, $n$ is the total number of time bins, $x_i$ is the location of the i-th time bin, $dt$ is the time bin duration, and $n_t$ is the number of spikes in the i-th time bin. Harris et al. [83] did not provide a value for $dt$, but a value equal to the sampling interval of the data will closely resemble the Leutgeb et al. [50,51] approach (Diamantaki et al. [84] use this configuration).

## Temporal KSDE

Brun et al. [56], Fyhn et al. [6] and Leutgeb et al. [57] describe a spatial mapping procedure which uses a KSDE approach but which averages the instantaneous firing rate of the cell in time, rather than treating all of the data within a bin as a homogenous temporal sample. Formally this approach can be described as:

$$\hat{f}(x) = \sum_{i=1}^{n} K(x_i - x)f_i \tag{17}$$

Here, $K$ is a weighting kernel, $n$ is the total number of time bins, $x_i$ is the location of the i-th time bin, and $f_i$ is the firing rate of the cell in the i-th time bin. To maintain the highest degree of accuracy we can choose $dt$, the sampling interval of the position data (i.e., 0.02s if sampled at 50Hz) as our time bin duration. We can then construct a spike train by calculating the number of spikes associated with every position data sample. The firing rate vector $f_i$ can then be calculated by convolving the spike train with a Blackman window of length $w \times s$ where $w$ is the desired duration of the window in seconds and $s$ is the sampling rate of the spike train. If this window has unit gain at zero frequency the resulting smoothed vector will sum to $n_s$, the total number of spikes, and dividing by $dt$ will give the instantaneous firing rate of the cell. Brun et al. [56], Fyhn et al. [6] and Leutgeb et al. [57] used a window duration of 2s, but through testing we found that using shorter duration windows increased the accuracy of the resulting maps (**S4 Fig**). We therefore used a window duration of 0.125 s (7 position data samples) as we wanted to maintain some temporal smoothing and accuracy was not greatly improved by using shorter windows.

Fyhn et al. [6] defined $K$ as a second Blackman kernel, but for simplicity we replaced this with a Gaussian. In simple terms, this approach calculates the instantaneous firing rate of the cell throughout the session, spatial firing rate is then estimated at a desired location by Gaussian weighting these values according to their proximity to the bin. Because this approach is based on the individual passes through a bin, rather than treating all samples within the bin as a single group, it is likely more sensitive to the temporal consistency of spatial firing.

## Map accuracy

For each rate map parameter combination we compared the resulting rate map $r(x,y)$ to the underlying PDF of the cell $f(x,y)$ using mean integrated squared error (MISE):

$$MISE = E \int_{x_0}^{x_t} \int_{y_0}^{y_t} (r(x,y) - f(x,y))^2 dxdy \qquad (18)$$

where $x_0$ and $x_t$ were the first and last bin edge in the x dimension and $y_0$ and $y_t$ were the same in the y dimension, $r(x,y)$ and $f(x,y)$ were both normalized to unity and $E$ represents the numerical average. Firing rate maps often contain missing data (i.e. unvisited) bins, which makes true integration over the maps impossible, so the integration was approximated by the Riemann sum over bin centers. For this reason $f(x,y)$ was evaluated at a precision of 1mm and $r(x,y)$ was interpolated to 1mm precision using nearest neighbor interpolation (MATLAB *interp2*), $dx$ and $dy$ were set to 1. Alternative error measures are explored in **S5 Fig**.

## Parameter optimization

The parameters we wanted to minimize were often in conflict, for example, it is generally not possible to reduce MISE without also increasing computation time. To tackle this issue, we used multiobjective optimization to find solutions which were Pareto-optimal, that is to say, solutions which cannot be improved without diminishing at least one of the objective functions [85].

To achieve this, we used an elitist genetic algorithm (MATLAB *gamultiobj*) which was given the MISE, computation time, proportion of missing data and place field detection error rate for every firing rate map parameter combination. The algorithm was restricted to between the smallest and largest bin size and smoothing values tested. The proportion of missing data was calculated as the number of empty bins (equal to *NaN*, not counting bins equal to zero) divided by the total number of bins. The place field detection error rate was calculated as the difference between the number of place fields detected in the firing rate map and the number detected in the actual spike probability map (sampled at 1 mm resolution). Place fields were detected as contiguous regions greater than 36 cm$^2$ remaining after thresholding the map at 20% of its maximum value.

## Pareto-optimal parameters

Multiple Pareto-optimal solutions can be found for a given data set. To choose among the returned solutions, we min-max normalized the Pareto-front solutions and then calculated their Euclidean distance, in terms of MISE and computation time, from the 'Utopia point' which was set to the minimum MISE and computation time found in the Pareto-front. In this way, our Pareto-optimal solution represents the best trade-off between error and computation time.

## Estimating field size

The equations we provide for estimating the ideal mapping parameters depend on a recording duration and firing field size. One method to estimate firing field size without a firing rate map

involves the Ripley's K-function [59,60] (**S6A Fig**). Formally, Ripley's K-function [59] represents the expected number of points $N$ within a distance $r$ of another point normalized by their density, $\lambda$:

$$K(r) = \frac{1}{n} \sum_{i=1}^{n} N_{p_i}(r)/\lambda \tag{19}$$

where $p_i$ is the current point, and the sum is taken over $n$ points. For comparison, the value of $K(r)$ for a random Poisson distribution is always equal to $\pi r^2$, thus deviations above or below this expected value indicate clustering and dispersion respectively. Besag [86] proposed a normalization for the K-function so that the expected value is always $r$:

$$L(r) = \sqrt{(K(r)/\pi)} \tag{20}$$

Subtracting $r$ from this value gives a function where deviations are above or below zero:

$$H(r) = L(r) - r \tag{21}$$

We then estimated firing field size as the radius associated with the maximum value of $H(r)$ multiplied by 0.75 (**S6**b Fig). Kiskowski [60] previously found that the maximum value of $H(r)$ approaches twice the clustering radius of simulated distributions, however, this is dependent on the spacing between clusters and because the distance between place fields is not restricted to 4× the field radius, as it was in Kiskowski [60], we found a smaller value to be more accurate.

## Bin size rules-of-thumb

We tested bin size rules-of-thumb and compared them to our quantitatively derived values. Sturge's rule [12] estimates the number of required histogram bins as equal to:

$$nbins = \lceil log_2 N + 1 \rceil \tag{22}$$

where $N$ is the total number of data points. We applied this rule to the total number of position data samples and utilized $nbins$ in both the x- and y-axis.

For a one-dimensional histogram the Freedman-Diaconis [14] rule calculates the number of bins along the x-axis as equal to:

$$nbins = 2IQR(x)N^{-1/3} \tag{23}$$

Where $x$ are the data values, IQR is the interquartile range, as before $N$ is the total number of data points. To extend this into two dimensions we used two approaches, in the first we estimated the number of bins in the x- and y-axis as equal to

$$nbins_d = \lceil (\max(x) - \min(x))/nbins \rceil \tag{24}$$

where $x$ are the data values along dimension $d$. We applied this to the x- and y-dimensions of the position data separately and took the maximum value. In the second approach we normalized the two-dimensional position data using the L2 norm:

$$\|v\| = \sqrt{x^2 + y^2} \tag{25}$$

Where $x$ and $y$ are the position data values in the x- and y-dimensions respectively and $\|v\|$ are the normalized, one-dimensional values. We then applied the Freedman-Diaconis [14] rule to these normalized values and created $nbins$ in both the x- and y-axis.

### Literature survey

To compare our results to the parameter combinations used in the published literature, we surveyed spatial navigation papers using Google Scholar and the PubMed database using related keywords such as 'place cell', 'grid cell', 'hippocampus' and 'rate map'. Relevant articles were scanned for information regarding their rate map approach and settings. Articles were excluded if the approach or bin size were not clearly specified. If a Gaussian smoothing kernel was used but the standard deviation was not specified (9 articles) a standard deviation equal to 1 bin was assumed. If boxcar smoothing was used but the kernel size was not specified (1 article) a kernel size of 3×3 was assumed. This information was categorized and accumulated, and all values were converted to mm. Surveying was halted when the total number of bivariate histogram entries (the most widely adopted approach) exceeded 100.

### Code availability

Code to generate firing rate maps using any of the methods described in this manuscript is available in the GitHub repository at https://github.com/RoddyMGrieves/rate_mapper.

### Supporting information

**S1 Fig. Histogram rate map errors when smoothing before or after division. A)** Example firing rate maps where smoothing was applied to the spike and dwell maps before division. **B)** Same as a but these maps were created by smoothing the rate map after division. Note the greater number of empty bins. **C)** MISE maps for the approaches in a and b respectively, shown as in **Fig 7**. d) Smoothing before division error map minus the smoothing after error map. Smoothing before division is less accurate at small bin sizes, the two approaches are comparable otherwise.
(TIF)

**S2 Fig. Adaptive implementations compared.** Four example cells, one per row. Columns show each cell's spike probability map, simulated spikes and trajectory, firing rate map generated using the pixelwise adaptive smoothing approach described by Skaggs and McNaughton [42], a convolution implementation of this, the pixelwise adaptive binning approach described by Yartsev and Ulanovsky [44], and a convolution implementation of this. Bar graphs show the average time taken to generate the maps in each column.
(TIF)

**S3 Fig. KSDE approaches compared.** Four example cells, one per row. Columns show each cell's spike probability map, simulated spikes and trajectory, and three firing rate maps generated using a MATLAB implementation of the kernel described by Leutgeb et al. [50,51], a pixelwise implementation of this (i.e. as described in the literature) and an implementation using the built-in MATLAB *mvksdensity* function and its kernel respectively. Bar graphs show the average time taken to generate the maps in each column. Because the last implementation was the fastest and did not visibly differ from the others, we used that approach.
(TIF)

**S4 Fig. Choosing a window duration for temporal KSDE. A)** Three example cells, one per row. Columns show firing rate maps generated using the temporal KSDE approach (bin size = 20 mm, spatial smoothing = 30 mm) with increasing temporal smoothing window durations. **B)** MISE maps for increasing temporal smoothing window durations. Bottom right text gives the minimum MISE value in each map. The color axis used here does not show the full

range of data values but was chosen to visually isolate any low error region(s).
(TIF)

**S5 Fig. Alternative map error measures were generally consistent with MISE. A)** Error plots for 64 cells simulated in 8-minute sessions for each mapping method (rows) and for multiple error measures (columns), from left to right: mean integrated squared error (MISE), pairwise Pearson correlation, Euclidean distance and Mutual Information (Matlab function *mi*, J. Delpiano). There is little difference between the measures. Note that plots are shown using a consistent color axis which may not span the full range of data values for every method. **B)** Top: the balanced mapping solution bin size for each mapping approach and error measure, normalized relative to MISE. Text above gives the result of a one-sample t-test comparing each group to zero (* = p < .05, n.s. = not significant). Plots below show the same but for the balanced solution smoothing strength, minimum solution bin size and minimum solution smoothing strength. For visualization, a small amount of jitter was added to the x-axis values in these plots.
(TIF)

**S6 Fig. Estimating field size without a rate map. A)** Left: The spikes and trajectory of an example simulated place cell. Overlaid are test radii of increasing size. Middle: for each radius, the total number of spikes that fall within its area (blue line) and the number of spikes we would expect if they were distributed uniformly across the animal's trajectory (black line). Right: the number of spikes minus the expected number and the maximum of this function. The maximum is related to the spatial clustering or place field size of the place cell. Ripley's K function is related to this example, but it is repeated across many different locations in the environment and the results are averaged. **B)** Left: the result of applying the Ripley's K procedure to all our simulated place cells, colors denote the three firing field size groups (Fig 1). Black line shows a linear fit to the data (Spearman's correlation between estimated and real field size: r = 0.71, $p = 3.6 \times 10^{-118}$). Gaussian jitter (mean = 0, σ = 8) was added to the y-values to make visualization of overlapping data clearer. Right: kernel smoothed probability density estimate of the data (no jitter; Matlab *ksdensity* with 25 mm bandwidth).
(TIF)

**S7 Fig. Overdispersion has little effect on the balanced or minimum error mapping solutions. A)** Two example cells, one per column. The left cell's spiking activity does not include overdispersion, the bottom plot shows the variability in the cell's firing during passes through the place field (black bars; text above gives the standard deviation of this distribution) which matches closely the expected distribution if the cell was modulated by space alone (red line). The right cell's spiking does include overdispersion. **B)** The MISE error plots for 64 cells simulated in 4-minute sessions without overdispersion (left column) and with overdispersion (right column) for each mapping method (rows). There is very little difference between the columns. Note that plots are shown using a consistent color axis which may not span the full range of data values for every method. **C)** Left: balanced and minimum error solution bin sizes did not change when overdispersion was included. Right: balanced and minimum error solution smoothing strength tended to be significantly higher when overdispersion was present. Text above gives the result of a one-sample t-test comparing each group to zero (* = p < .05, n.s. = not significant). **D)** The MISE (left plot) and error in detecting place fields (right) did not differ when overdispersion was present if using the balanced solution. For visualization, a small amount of jitter was added to the place field plot's x-axis values.
(TIF)

**S8 Fig. Sampling bias has no effect on the balanced or minimum error mapping solutions. A)** Three example trajectories, 8-minutes long, one per column, from left to right: uniform sampling of the environment, a strong bias to one location such as a goal and a bias for moving close to the walls. Plots below show the dwell time map for each trajectory. **B)** The MISE error plots for 64 cells simulated in 8-minute sessions, for each of the trajectory types (columns) and for each mapping method (rows). There is very little difference between the columns. Note that plots are shown using a consistent color axis which may not span the full range of data values for every method. **C)** Left: balanced and minimum error solution bin sizes did not change when trajectories were biased. Values are normalized relative to the 'uniform' trajectory results. Text above gives the result of a one-sample t-test comparing each group to zero (* = p < .05, n.s. = not significant). Right: same for smoothing strength. **D)** The MISE (left plot) and error in detecting place fields (right) when using the balanced solution for each trajectory type. MISE increased for the 'goal' trajectory, which can also be seen in the middle column of panel b, which is darker than the others. For visualization, a small amount of jitter was added to the place field plot's x-axis values.
(TIF)

**S9 Fig. Mapping parameters applied to real place cells.** Top row shows the spike (red dots) and position data (black line) for 5 place cells from the dataset of [20] available online. Below each of these, firing rate maps are shown that were generated using the balanced solution parameter equations shown in Table 1 (left column of each cell) and maps generated using the minimum error solution parameter equations shown in Table 1 (right column of each cell). In both cases parameters were calculated assuming an average field radius of 300 mm and a recording duration of 25 mins. One mapping method is shown per row.
(TIF)

**S10 Fig. Mapping parameters in the literature result in very different rate maps.** Related to Fig 11B. Rows show the results for the Histogram, Adaptive smoothing and KSDE mapping methods respectively. Left column: MISE map overlaid with the minimum error, balanced and literature reported parameter combinations as in Fig 11B. Gaussian jitter (mean = 0, σ = 2) was added to the literature values to make visualization of overlapping data clearer. For each mapping method 3–5 literature values have been selected and numbered at points of interest. Middle column: the MISE associated with the minimum error and balanced parameter combinations and the numbered points in the MISE map for all 256 simulated place cells. In each plot groups differed significantly (Histogram: $F(6,1785) = 167.9$, $p = 1.93 \times 10^{-169}$; Adaptive: $F(5,1530) = 73.6$, $p = 3.36 \times 10^{-69}$; KSDE: $F(4,1275) = 74.5$, $p = 8.06 \times 10^{-57}$; one-way ANOVA) and horizontal lines denote a significant ($p < .05$) post-hoc comparison. MISE values vary greatly across the different literature values and the numbered points are generally less accurate than the balanced solution. Right column: firing rate maps generated for an example place cell using the minimum error and balanced parameter combinations and the numbered points in the MISE map. While some parameter combinations produce maps similar to the minimum error or balanced solution, they often deviate significantly.
(TIF)

**S11 Fig. Mapping parameters in the literature are generally accurate for place field detection.** Related to Figs 11B and **S10**. Rows show the results for the Histogram, Adaptive smoothing and KSDE mapping methods respectively. Left column: place field detection error map overlaid with the minimum error, balanced and literature reported parameter combinations as in Fig 11B. Gaussian jitter (mean = 0, σ = 2) was added to the literature values to make visualization of overlapping data clearer. For each mapping method 3–5 literature values have been

selected and numbered at points of interest. Right columns: the field detection error, computation time and proportion of empty bins respectively associated with the minimum error and balanced parameter combinations and the numbered points for all 256 simulated place cells. Horizontal lines denote a significant (p < .05) post-hoc comparison following a significant Kruskal-Wallis omnibus test.
(TIF)

**S1 Data. Excel spreadsheet containing, in separate sheets, the underlying numerical data and statistical analysis for Fig panels 9A-9F, 10A-10D, 11A–11B, S5B, S6B, S7C–S7D, S8C–S8D, S10 and S11.**
(XLSX)

## Acknowledgments

The author would like to thank Éléonore Duvelle for her comments and advice on this manuscript.

## Author Contributions

**Conceptualization:** Roddy M. Grieves.

**Data curation:** Roddy M. Grieves.

**Formal analysis:** Roddy M. Grieves.

**Investigation:** Roddy M. Grieves.

**Methodology:** Roddy M. Grieves.

**Software:** Roddy M. Grieves.

**Visualization:** Roddy M. Grieves.

**Writing – original draft:** Roddy M. Grieves.

**Writing – review & editing:** Roddy M. Grieves.

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
