## [Decision Letter · Decision Letter 0]

10 Sep 2023

Dear Dr. Grieves,

Thank you very much for submitting your manuscript "Estimating neuronal firing density: a quantitative analysis of firing rate map algorithms" for consideration at PLOS Computational Biology.

As with all papers reviewed by the journal, your manuscript was reviewed by members of the editorial board and by several independent reviewers. In light of the reviews (below this email), we would like to invite the resubmission of a significantly-revised version that takes into account the reviewers' comments.

In particular, myself and the reviewers believe that this will be an important and useful resource for the field, which puts the process of generating firing rate maps from single cell recordings on a firmer theoretical footing. However, the manuscript would benefit from a little more rigorous statistical analysis (as highlighted by Reviewers 1 and 3), as well as some further exploration of how these methods cope with more realistic data sets (i.e. without good / even coverage of an environment) and / or firing fields that exhibit overdispersion.

We cannot make any decision about publication until we have seen the revised manuscript and your response to the reviewers' comments. Your revised manuscript is also likely to be sent to reviewers for further evaluation.

Sincerely,

Daniel Bush

Academic Editor

PLOS Computational Biology

Thomas Serre

Section Editor

PLOS Computational Biology

I share the reviewer's opinion that this will be an important and useful resource for the field, which puts the process of generating firing rate maps from single cell recordings on a firmer theoretical footing. However, the manuscript would benefit from a little more rigorous statistical analysis (as highlighted by Reviewers 1 and 3), as well as some further exploration of how these methods cope with more realistic data sets (i.e. without good / even coverage of an environment) and / or firing fields that exhibit overdispersion.

Reviewer's Responses to Questions

**Comments to the Authors:**

Reviewer #1: ## Title

Estimating neuronal firing density: a quantitative analysis of firing rate map algorithms

## Summary

Six different algorithms for binning up spatial (x,y) position and spiking data to create firing rate maps are evaluated and compared to a "ground truth" data set where the spiking pdf is known and to each other in terms of computation time, error (cf ground truth) and accuracy. The conclusion reached is that the bivariate histogram (which remains the most commonly used method for the last 50 years), plus some smoothing, is the optimal solution almost all of the time.

It's nice to see this kind of analysis collected in one place as the effects of various parameters on binning data are often overlooked in the literature, and if they are mentioned, they are usually buried across several papers. However, I have several concerns I would like to see addressed before recommending for publication.

## Main points

One of the main difficulties in assessing the algorithms reported here is that as of this writing (01/08/23) the code was still not publicly available on Github (although I applaud the authors efforts to publish it openly eventually).

Although the paper claims to be a quantitative assessment there is not a single statistical test conducted on any of the results. Graphing out the performance of the various methods is fine and gives the reader a qualitative sense of how well they do over a range of values but a quantitative comparison would be useful too.

It is stated in the Introduction that the experimental sampling of position is often non-uniform. The position data generated here is fairly close to uniform, a desired end-state in many open-field recordings. However, some of the binning methods presented here are explicitly designed to deal with cases where sampling of one or more variables is non-uniform. It would therefore be useful to see how the different binning methods deal with positional data that is non-uniform i.e. where there are chronically *under-sampled* areas of the simulated environment.

## Specific points

1. Abstract - line 24 - "The analysis of spatial neurons...", surely all neurons are spatial? Be more specific with your language - "neurons that exhibit receptive fields dependent on an organisms spatial location" or similar.

2. Somewhere in the Introduction (around line 91 would be a good place) a list of the different mapping techniques should be provided

3. You mention that Sturges rule is inappropriate for non-normal data, yet use it throughout the rest of the paper. Justify this.

4. Simulated dataset - line 126 - are the generated trajectories actually unique? Looking closely at Figure 1E it looks like the shorter length trajectories are actually subsets of the longer ones.

5. Figure 1. It's unclear at this point what "variance in x and y" means (you have to dig into the text/ methods to find) and so it isn't really necessary - I think it defines how skewed the receptive fields are.

6. Line 183 - you mention that the smoothing kernel used is Gaussian - there are other smoothing kernels, why use a Gaussian and not another? It needs justification in light of the authors statement in the Introduction that the shape of place fields are rarely normally distributed.

7. This is a personal preference for the order in which the Figures are presented but you present schematics of the histogram method then one concerning a qualitative assessment of the different methods, then figures, in order, of the adaptive smoothing and then KSDE methods. Surely it's more logical to present each of the methods in order and *then* the figure showing the qualitative assessment of the different methods?

8. Line 201-2 - makes it sound like smoothing has occurred before *and* after. Tighten the language up.

9. Line 224 - "One issue that has been raised..." reference this immediately after.

10. Line 241 - any speculation as to why this has never been implemented?

11. You use the same spike/ position data for Figures 2 & 5 but not for the other examples of how the binning methods work. It would be nice to have the same spiking / position data for all of them to facilitate an comparison with the eye of truth.

12. The introduction to adaptive smoothing (line 306+) diverts into talking about 3-dimensional binning which is off-topic. You should limit the discussion to papers binning up 2D data. Despite this, I found the explanation unclear as to what the result of the binning process is. It sounds as though you end up with bins of unequal size, which would make comparison between maps with different positional sampling impossible. An explanation of how data is allocated to a bin should be more explicit i.e. what you end up with is a regularly spaced, but unevenly sampled, grid.

13. Line 328 "...though..." should read "through".

14. Line 340 "...the most widely adopted approach..." by whom? Provide references.

15. Line 357 "...firing rate map" should more accurately read "discretized map"

16. Fig 8. Returning to a point made earlier, there is no comparison being conducted here beyond a visual one. This figure is a good example - correlations could be performed between the 3 different firing rate map types produced and an ANOVA could be conducted on the computation time.

17. Lines 433-45 I'm not sure how the bin sizes were calculated here; my calculation of Sturges rule gives bin sizes of 17 and 19mm for 16 and 64 minute recordings respectively which intuitively feels right as opposed to the 70 and 63mm sizes given in the text (these are pretty large bins given the usual 2-3cm employed in most experimental papers). Similarly, my calculation of the Friedman-Diaconis rule didn't agree (I got 33 and 20mm). I also feel as though there was no rationale provided as to why one rule of thumb was used over another. Please provide one.

18. State why an "abundance of empty bins is undesirable".

19. Figure 10 I'm slightly unclear what some of the numbers mean. What does an error of 0.5 mean? 0.5%? 50%? Or something else? The same applies to the "proportion of map empty" plots.

20. Do the values in Table 1 need to be accurate to 3 decimal places? Seems like overkill

21. Figure 12 - it wasn't clear where/ how the centre of the circles were placed.

22. I'm unclear on the preoccupation with the proportion of empty bins (especially given that the positional sampling is almost perfectly uniform) cf Figure 13d

23. Figure 13e - needs colour bar

24. Line 628 - "a weak, but significant, decrease..." no statistics are reported to back this statement up

25. I don't think it's necessarily that "the field has recognised the increased accuracy afforded by smoothing" that has led to the adoption of Gaussian smoothing kernels. If you have an a priori reason that the underlying response function is Gaussian (and we do with place fields for eg) then the raw data should be smoothed with a kernel that reflects that assumption, thus emphasising the underlying response and de-emphasising the things you don't want.

26. Line 661 "balance" should read "balanced"

Reviewer #2: In the current manuscript, Grieves provides a thorough analysis of several methods for quantifying position-selective firing of recorded neural data. I find the work to be detailed and well-described. Despite the fact that this is a rather mature field, I am unaware of a similar study which directly compares place field quantification methods and criteria, making this a novel and important study.

I have only one concern, which regards the manner in which spiking was generated from the ground truth place fields. The author uses a Poisson distribution to generate spike patterns. However, Fenton & Muller (PNAS, 1998) showed that firing of place cells is rarely well described by a Poisson distribution. Rather, individual passes through place fields display “overdispersion” (at least, in some cases and/or environments). Overdispersion makes quantifying place fields particularly challenging for cells with relatively low max firing rates or for experiments with limited behavioral coverage of the environment.

Thus, I would like the author to repeat some of the core analyses using spikes generated from an overdispersion model (see Olypher et al, Neuroscience, 2002). I don’t think it’s necessary to repeat all analyses, only those where the variability introduced by overdispersion would be most impactful: low peak firing rate place fields and short-duration recordings.

An additional minor critique is that the author uses the term "we" throughout the manuscript even though it is a one-author paper. I believe this should be changed to "I" or rephrased to remove the subject pronoun entirely from these sentences.

Reviewer #3: This manuscript reviews existing and new methods and parameters used to generate firing rate maps of spatially modulated neural activity. In particular, the author thoroughly explores the impact of those methods and choice of parameters on how the rate maps correspond to the underlying spike distribution, which is derived from simulated data.

Overall, the analysis is rigorously performed, and the results convincingly demonstrated. The comparisons performed in the paper are informative for the spatial cognition field. However, I have concerns about the novelty and the impact of the results of this paper. In terms of novelty, the author showed that the bivariate histogram with spatial smoothing was the preferred choice in the majority of cases, but this is already the most commonly used method to compute rate maps. Similarly, in terms of impact, the author shows a method to optimise the parameters used when computing rate maps, but those optimal parameters are in the range most commonly used in the literature. Furthermore, the fact that this study is only performed on synthetic data makes it difficult to gauge the efficiency of this method on real datasets.

Major:

- The way the author quantifies the rate map error is difficult to reconcile with a tangible measure. It would be helpful to provide other measures of similarity (correlation, mutual info, population vector distance, …) between the ‘true’ rate map and the computed rate maps. In its present form, the MISE is adequate to investigate the point that the author is making (eg: best method and parameters for rate map computation) but it is difficult to relate the MISE and thus the variation of method and parameters to concrete and more classic similarity measures. The author uses the place field detection, compute time, and proportion of empty bins, but adding other measures, classically used in the literature would strengthen the message and scope of the method detailed in this paper. In its present form, I am circumspect about the impact of these optimisations on a real dataset.

- This analysis is performed on synthetic data, where the author fitted as much as possible various parameters of real datasets (see methods). However I am concerned about the robustness of these results on real datasets. Notably, with different place field size (more variation than presented in this paper), different geometry of the environment (linear/star mazes, multiple rooms, very large environments....). The demonstration of the best method to bin spike data needs a synthetic dataset in order to access the underlying spike probability. However, to optimise for the detection of place fields, compute time, and empty bins (fig 10-13), in some cases, the author does not necessarily need the spike probability distribution. In these cases, it would be interesting and easy for the reader to grasp the effect of the parameter optimization on a real word dataset. In a real dataset, the place field size, number, firing rate and distribution will vary much more than for the simulated place cell population of this paper ( random distribution of place fields , same firing rate, different place field size in a single cell,... ). All these discrepancies, notably the difference in firing rate and size of place fields have the potential to strongly influence the optimal parameter choice (optimal smoothing,... ).

- 46 “These parameters can have a huge impact on the map created”, The author shows that this is the case in theory. However, the author shows in Fig 14 that for previous work the solution is very close to the balanced solution. I thus wonder how big is the difference in place field detection, MISE,... in the range of parameters classically used in the field. According to figure 13, the difference might be small.

Minor:

- The authors make no mention of the padding used when smoothing. This can have a drastic impact on the calculated error, especially for large half windows. It could also have a different influence on spike distribution with high spike probability in the centre or at the periphery of the environment

- Line 163, the authors should also consider citing the following papers which are to my knowledge the first published report of rate maps (although in radial mazes).

McNaughton, B.L., Barnes, C.A. & O'Keefe, J. The contributions of position, direction, and velocity to single unit activity in the hippocampus of freely-moving rats. Exp Brain Res 52, 41–49 (1983).

Barnes, Carol A., Bruce L. McNaughton and John O’Keefe. “Loss of place specificity in hippocampal complex spike cells of senescent rat.” Neurobiology of Aging 4 (1983): 113-119.

- 461 “Generally, the error in place field detection was anti-correlated with map error (Fig 10; i.e., histogram) but this was not always the case (i.e., ASH).” This is not shown directly in the figure, the reader needs to integrate the results of figure 9 and 10 to deduce this. The author should show it in a figure

- 557: figure 12I does not exist, I think the author cite 12b instead

- In figure 12b the author does not comment on the fact that the estimated field radius with Ripley k is bimodal, notably for place field size under 250mm. This bimodality could influence the correlation reported in the figure.

- Figure 13: missing legend for e)

**Have the authors made all data and (if applicable) computational code underlying the findings in their manuscript fully available?**

Reviewer #1: **No: **The Github repository that contains the code is not public as stated in my review above

Reviewer #2: Yes

Reviewer #3: Yes

PLOS authors have the option to publish the peer review history of their article (what does this mean?). If published, this will include your full peer review and any attached files.

Reviewer #1: No

Reviewer #2: No

Reviewer #3: No
---

## [Decision Letter · Decision Letter 1]

27 Nov 2023

Dear Dr. Grieves,

Thank you very much for submitting your manuscript "Estimating neuronal firing density: a quantitative analysis of firing rate map algorithms" for consideration at PLOS Computational Biology. As with all papers reviewed by the journal, your manuscript was reviewed by members of the editorial board and by several independent reviewers. The reviewers appreciated the attention to an important topic. Based on the reviews, we are likely to accept this manuscript for publication, providing that you modify the manuscript according to the review recommendations.

In particular, a few details of the comparisons between different rate map methods still need to be clarified and justified, in line with comments from Reviewer #3, before this manuscript can be accepted for publication.

Sincerely,

Daniel Bush

Academic Editor

PLOS Computational Biology

Thomas Serre

Section Editor

PLOS Computational Biology

Reviewer's Responses to Questions

**Comments to the Authors:**

Reviewer #1: I thank the authors for making the amendments to the MS and making the data and code-base available.

Reviewer #2: The authors have fully addressed my original concerns.

Reviewer #3: This manuscript represents an updated version of a prior submission, offering a comprehensive review of both traditional and novel techniques for creating firing rate maps of spatially modulated neurons. The author has extensively investigated how these methods and their parameter choices align the computed firing rate maps with the underlying, simulated, spike distribution data. Building on the original manuscript, the author has conducted detailed analyses and compiled a valuable summary of various approaches for producing firing rate maps of spatially modulated neurons. This revised version also addresses several minor issues raised earlier and includes additional insightful analysis. Despite these enhancements, I still have reservations regarding the paper's originality, practical value, and overall impact. One of the main results of this paper is that the bivariate histogram method, combined with spatial smoothing, has been identified as the preferred method in the majority of scenarios. However this method is already widely acknowledged as the simplest and most commonly used approach for constructing rate maps. The parameter optimization could also lack a “real word” use case in my opinion. The scale of the differences reported in the paper (computation time, MISE error,...) is really small, and the condition of this demonstration is very (too?) controlled and is not shown to hold with the variety of arena shape and scale but also the variety of place field size and other factors as mixed selectivity for example.

Overall, the principal merit of this paper lies in its methodical compilation and examination of various methods for calculating rate maps, alongside a review of relevant literature in the field of spatial cognition. This comprehensive approach makes the paper a valuable resource for researchers in the field. However, I maintain reservations regarding the strength of its central message. The differences it reports are quite minimal, even using synthetic data. Consequently, I remain sceptical about the practical application and effectiveness of the parameter optimization strategies proposed in this paper in real-world scenarios.

I will first answer the response to the previously review then add other minor comments to the updated manuscript:

Answers to main points:

Answer to main point 1:

Despite the MISE being “the primary measure used to quantify and optimize density estimation techniques since the 1950s” I think it still fails to help the reader to grasp the extent of the difference between the different parameters/methods studied here. The MISE being in the range of 10-12 at worst (see fig S5) means that the average difference between the observed and synthetic rate map is ~10-6 Hz. This difference is very small and unlikely to make a significant difference in real, less controlled dataset. This lack of consequential effect of the parameter optimization is reflected by the difference (best vs worst case) of correlation: +- 0.1 , euclidian +- 0.07 Hz . This method and parameter changes make a difference on synthetic data (place field detection,....) but are unlikely to make a significant difference on a real dataset with more noise (SNR, non gaussian field, inhomogeneous sampling, environment shape,...). For Mutual Information, a quick test seems to show the calculation of MI should be barely longer than calculating the correlation between two vectors (with a size comparable to the ones studied here). Concerning the population vector distance or any population vector analysis suggested previously, storing rate maps should be a problem with modern computers. Storing ~1k rate maps with the maximum binning size of this study should not even take a gigabyte of RAM (in float64).

Answer to main point 2:

This answer illustrates some of the previous criticisms I had on the lack of robust methods to grasp the real advantage of the methods and parameters on real data. Effectively, real data will not allow access to the underlying spike probability distribution and thus make the comparison with MISE impossible. However, the other comparison possibilities suggested by the author are not satisfactory as they heavily depend on the machine used for the analysis and the code implementation (language, CPU/GPU, various optimisations,...). To complement the additional illustration shown in figure S9, the author could quantify the error of place field detection, the MISE or an other distance metric between the worst parameter choice reported in the literature and the minimum or balanced parameter.

Answer to main point 3:

Effectively, it would be interesting to correlate the MISE error (Fig 11b) and the variable compared in Fig 8. I encourage the author to show such figures. With an indirect comparison, the colormap does not allow an accurate comparison. Furthermore, as the author highlights, only a handful of studies chose parameter falling in the high error area and if so these points seems to fall in low place field detection error (max ~15%, with the bivariate histogram, it might be less but the colormap does not allow to distinguish).

Answer to minor points. The points not cited here do not need further discussion. I thank the author for his response to these points:

Answer to minor point 1:

The use of zero padding “to reduce edge effects” is incorrect. Zero padding rate maps with non-zero out of field firing (most if not all spatial rate maps) will introduce a sharp transition at the location of the edge. This might not be a problem on synthetic data, but in my experience the mirroring or reflect option is preferable. No option will be a perfect/correct option as the data is not recorded outside of the arena but the phrasing is incorrect. I highly encourage the author to use another padding method and if not to rephrase the modification (line 850, 909, 952) saying that padding removes edge effect.

Answer to minor point 5:

The term "satisfactory" does not describe objectively a correlation between two variables

Furthermore, "mode somewhat separated": the author could test his hypothesis - that the bimodality is caused by cut field - easily by selecting cells with place fields at the centre of the maze.

Minor comments

The figure linked to the last version of the file are extremely blurry

The text in some figure is very small and could easily be increased to improve readability (eg: in Fig 2: text in the top left of the environments)

**Have the authors made all data and (if applicable) computational code underlying the findings in their manuscript fully available?**

Reviewer #1: Yes

Reviewer #2: Yes

Reviewer #3: Yes

PLOS authors have the option to publish the peer review history of their article (what does this mean?). If published, this will include your full peer review and any attached files.

Reviewer #1: No

Reviewer #2: No

Reviewer #3: No

Figure Files:

Data Requirements:

Reproducibility:

References:

---

## [Decision Letter · Decision Letter 2]

15 Dec 2023

Dear Dr. Grieves,

We are pleased to inform you that your manuscript 'Estimating neuronal firing density: a quantitative analysis of firing rate map algorithms' has been provisionally accepted for publication in PLOS Computational Biology.

Best regards,

Daniel Bush

Academic Editor

PLOS Computational Biology

Thomas Serre

Section Editor

PLOS Computational Biology

Reviewer's Responses to Questions

**Comments to the Authors:**

Reviewer #3: I thank the author for the answers he provided to my concerns and his modifications to the manuscript

**Have the authors made all data and (if applicable) computational code underlying the findings in their manuscript fully available?**

Reviewer #3: Yes

PLOS authors have the option to publish the peer review history of their article (what does this mean?). If published, this will include your full peer review and any attached files.

Reviewer #3: No

---

## [Editor Report · Acceptance letter]

19 Dec 2023

PCOMPBIOL-D-23-00912R2 

Estimating neuronal firing density: a quantitative analysis of firing rate map algorithms

Dear Dr Grieves,

I am pleased to inform you that your manuscript has been formally accepted for publication in PLOS Computational Biology. Your manuscript is now with our production department and you will be notified of the publication date in due course.

With kind regards,

Judit Kozma
